# RFWD3 and translesion DNA polymerases contribute to PCNA modification–dependent DNA damage tolerance

Rie Kanao[1,2], Hidehiko Kawai[3], Toshiyasu Taniguchi[4], Minoru Takata[5], Chikahide Masutani[1,2]

**DNA damage tolerance pathways are regulated by proliferating cell nuclear antigen (PCNA) modifications at lysine 164. Translesion DNA synthesis by DNA polymerase $\eta$ (Pol$\eta$) is well studied, but less is known about Pol$\eta$-independent mechanisms. Illudin S and its derivatives induce alkyl DNA adducts, which are repaired by transcription-coupled nucleotide excision repair (TC-NER). We demonstrate that in addition to TC-NER, PCNA modification at K164 plays an essential role in cellular resistance to these compounds by overcoming replication blockages, with no requirement for Pol$\eta$. Pol$\kappa$ and RING finger and WD repeat domain 3 (RFWD3) contribute to tolerance, and are both dependent on PCNA modifications. Although RFWD3 is a FANC protein, we demonstrate that it plays a role in DNA damage tolerance independent of the FANC pathway. Finally, we demonstrate that RFWD3-mediated cellular survival after UV irradiation is dependent on PCNA modifications but is independent of Pol$\eta$. Thus, RFWD3 contributes to PCNA modification–dependent DNA damage tolerance in addition to translesion DNA polymerases.**

## Introduction

DNA damage tolerance prevents replication arrest at DNA lesions, and modifications of proliferating cell nuclear antigen (PCNA) on lysine 164 (K164) play important roles in controlling DNA damage tolerance pathways (Kanao & Masutani, 2017). RAD6–RAD18, an E2–E3 complex, mediates PCNA mono-ubiquitination in response to replication arrest in eukaryotic cells (Hoege et al, 2002). Mono-ubiquitinated PCNA preferentially interacts with Y-family DNA polymerases (DNA polymerase $\eta$ [Pol$\eta$], DNA polymerase $\iota$ [Pol$\iota$], DNA polymerase $\kappa$ [Pol$\kappa$], and REV1) via their ubiquitin-binding and PCNA-interacting domains to mediate translesion DNA synthesis (TLS) across DNA lesions (Kannouche et al, 2004; Bienko et al, 2005; Plosky et al, 2006; Wood et al, 2007). Pol$\eta$ plays a principal role in bypassing UV-induced DNA lesions. Furthermore, Pol$\eta$ can accurately bypass *cis–syn* thymine–thymine cyclobutane pyrimidine dimer (CPD) in vitro (Masutani et al, 2000) and suppress UV-induced mutagenesis in mammalian cells (Kanao et al, 2015b). Pol$\eta$ also bypasses cisplatin-induced intrastrand crosslink lesions in vitro (Masutani et al, 2000; Vaisman et al, 2000), and Pol$\eta$-deficient human cells show greater sensitivity to cisplatin treatment than Pol$\eta$-complemented cell lines or Pol$\eta$-proficient fibroblasts (Albertella et al, 2005). Replacing PCNA with exogenous PCNA mutated at K164 results in extreme sensitivity to UV-irradiation and cisplatin treatment, primarily because Pol$\eta$ inactivation (Kanao et al, 2015a). Y-family polymerases have cognate DNA lesions and are selected by protein–protein interactions with factors including PCNA (Powers & Washington, 2018). However, the requirement for ubiquitination is complicated. For instance, in TLS by Pol$\kappa$, PCNA ubiquitination is absolutely required to bypass benzo[$a$]pyrene-induced lesions, but is only partially required to bypass methyl methane sulfonate-induced lesions (Wit et al, 2015). We previously showed that replacing PCNA with mutant PCNA increases UV sensitivity in Pol$\eta$-deficient cells, indicating that PCNA modifications at K164 are required for DNA damage tolerance other than Pol$\eta$-mediated TLS, although the contribution to UV-induced DNA lesions is limited (Kanao et al, 2015a). K63-linked poly-ubiquitination of PCNA, mediated by E2-E3 complexes Rad6-Rad18 and Ubc13-Mms2-Rad5, promotes error-free DNA damage tolerance (Hoege et al, 2002). The error-free pathway uses the nascently synthesized sister chromatid as template DNA. Although the precise mechanism is unclear, fork reversal and template switching have been proposed (Pilzecker et al, 2019). In human cells, RAD51-dependent fork reversal has been observed after treatment with several DNA damaging agents (Zellweger et al, 2015). PCNA poly-ubiquitination is also required for fork reversal (Vujanovic et al, 2017). However, specific details of the template switching mechanism remain under investigation in mammalian cells.

Illudin S is a natural sesquiterpene compound isolated from mushrooms that has potential as an anticancer drug because of its toxicity against various types of tumor cells, including multi-drug-resistant cancer cells. However, its high toxicity in normal cells is an obstacle to therapeutic use (Kelner et al, 1997). To decrease toxicity

[1]Department of Genome Dynamics, Research Institute of Environmental Medicine, Nagoya University, Nagoya, Japan    [2]Department of Molecular Pharmaco-Biology, Nagoya University Graduate School of Medicine, Nagoya, Japan    [3]Department of Nucleic Acids Biochemistry, Graduate School of Biomedical and Health Sciences, Hiroshima University, Hiroshima, Japan    [4]Department of Molecular Life Science, Tokai University School of Medicine, Isehara, Japan    [5]Laboratory of DNA Damage Signaling, Department of Late Effects Studies, Radiation Biology Center, Graduate School of Biostudies, Kyoto University, Kyoto, Japan

Correspondence: masutani@riem.nagoya-u.ac.jp

and improve tumor selectivity, (−)-acylfulvene (AF), a semi-synthetic derivative of illudin S, and analogs, including (−)-6-hydroxymethylacylfulvene (irofulven), were developed (Woynarowski et al, 1997; McMorris, 1999; Woynarowska et al, 2000). Reduced forms of AF selectively produce alkylated purine bases at the 3- and 7-positions, resulting in the bulky DNA adducts 3-AF-adenine and 7-AF-guanine (Pietsch et al, 2011). In addition, 3-illudin S–adenine is formed in a cell-free reaction between illudin S and DNA (Pietsch et al, 2013). Thus, illudin S and its derivatives can behave as alkylating agents. The formation of AF-induced DNA adducts correlates with the expression of prostaglandin reductase (PTGR1), an enzyme that can reduce illudin S and its derivatives (Dick et al, 2004; Pietsch et al, 2013). Illudin S can also be reduced non-enzymatically and can react with DNA under physiological conditions (McMorris et al, 1990). Irofulven does not induce DNA–protein or DNA interstrand crosslinks (Woynarowski et al, 1997). Using calf thymus DNA, Gong et al (2007) showed that AF–DNA adducts were depurinated in phosphate buffer (pH 7.4) at 37°C in vitro (Gong et al, 2007). The major pathway required for the repair of abasic sites is base excision repair. However, Jaspers et al (2002) reported that x-ray cross complementation group 1 (XRCC1)-mutant CHO cells, which are deficient in base excision repair, are as resistant to illudin S as parental CHO cells, suggesting that abasic sites are not the major cause of the cytotoxicity of these compounds (Jaspers et al, 2002).

Previous studies evaluated the importance of nucleotide excision repair (NER) in protecting cells against illudin S and irofulven. NER occurs via two sub-pathways: global genome (GG)-NER and transcription-coupled (TC)-NER. TC-NER–deficient cells are significantly more sensitive to illudin S and irofulven than wild-type cells (Jaspers et al, 2002; Koeppel et al, 2004; Schwertman et al, 2012), suggesting that DNA lesions induced by these compounds disturb transcription and are repaired by TC-NER. However, cells that are GG-NER–deficient but TC-NER–proficient are only as sensitive to these compounds as wild-type cells, suggesting that DNA lesions are ignored by GG-NER (Kelner et al, 1994; Jaspers et al, 2002; Koeppel et al, 2004) and remain in the global genome. The stable analog 3-deaza-3-methoxynaphthylethyl-adenosine (3d-Napht-A) is a model adduct of illudin S or its derivatives, and an in vitro study showed that purified yeast RNA polymerase II (Pol II) stalls at 3d-Napht-A in template DNA (Malvezzi et al, 2017b), consistent with the idea that stalled Pol II initiates TC-NER to remove DNA lesions.

Illudin S impedes not only transcription, but also DNA replication, causing cell cycle arrest in the G1/S-phase (Kelner et al, 1987). In vitro, 3d-Napht-A blocks nucleotide incorporation by DNA polymerase α (Malvezzi et al, 2017a). In addition, RAD18 KO in chicken DT40 cells increases sensitivity to illudin S, suggesting that PCNA ubiquitination–dependent DNA damage tolerance pathways are involved in responses to illudin S (Jaspers et al, 2002). However, xeroderma pigmentosum (XP) group variant (XP-V) cells, in which Polη is inactivated (Johnson et al, 1999; Masutani et al, 1999), are not sensitive to illudin S, suggesting that Polη is not involved in bypassing illudin S–induced DNA lesions (Jaspers et al, 2002).

RING finger and WD repeat domain 3 (RFWD3) was first identified as a substrate of ataxia telangiectasia mutated (ATM)/ATM and rad3-related (ATR) (Matsuoka et al, 2007). RFWD3 forms a complex with murine double minute 2 (Mdm2) and positively regulates p53

(Fu et al, 2010). In addition, RFWD3 interacts with replication protein A (RPA) (Gong & Chen, 2011; Liu et al, 2011) and promotes its ubiquitination (Elia et al, 2015; Inano et al, 2017), which is required for homologous recombination (HR) at stalled replication forks (Elia et al, 2015). Recently, a patient was reported with a typical Fanconi anemia phenotype with heterozygous mutations in *RFWD3* (Knies et al, 2017). RPA and RAD51 recombinase ubiquitination by RFWD3 is required for repair of DNA interstrand crosslinks (ICLs) mediated by Fanconi anemia complementation group (FANC) proteins, which remove lesions from damage sites and allow HR (Inano et al, 2017). On the other hand, Gallina and colleagues showed that RFWD3 conducts error-prone TLS across DNA-protein crosslink, CPD, and ICL in vitro systems containing *Xenopus* egg extract. They also suggested that RFWD3 contributes to PCNA ubiquitination in vitro and in human cells (Gallina et al, 2021).

In mammalian cells, Polη-mediated TLS is the best-recognized mechanism of DNA damage tolerance that depends on PCNA modifications at K164. Other pathways are less understood, although PCNA ubiquitination plays multiple important roles (Cipolla et al, 2016; Leung et al, 2018). To understand PCNA modification–dependent and Polη-independent DNA damage tolerance pathways, we examined the effects of illudin S and irofulven. We found that PCNA modifications at K164 are required for overcoming replication blockage induced by these compounds. In addition, we found that human Polκ and RFWD3 contribute to overcoming replication arrest dependently on PCNA modification at K164. We demonstrated that RFWD3 plays an important role in overcoming replication blockage independent of the FANC pathway. RFWD3 has a role in tolerating UV damage that is dependent on PCNA modification, but independent of Polη. We conclude that RFWD3 is required for PCNA modification–dependent DNA damage tolerance.

# Results

### PCNA modifications at K164 are required to protect human cells against illudin S and irofulven

To examine PCNA modification–dependent and Polη-independent DNA damage tolerance pathways, we searched for DNA-damaging agents to which cells deficient in PCNA modification are sensitive but Polη-deficient cells are not. Previously, we established the SV40-transformed human fibroblast line WI38VA13, which expresses either wild-type or K164R (KR) mutant exogenous PCNA (Kanao et al, 2015a). Exogenous PCNA has silent mutations in its nucleotide sequence at the target site of an siRNA against endogenous PCNA, which allows replacement of endogenous PCNA with exogenous PCNA via siRNA knockdown. In this study, we refer to the PCNA-replaced cells as PCNA[WT] or PCNA[KR]. In PCNA[KR] cells, mono-ubiquitinated PCNA was rarely detected, even after UV irradiation (Fig S1A). Sensitivity of the PCNA[KR] cells to mitomycin C (MMC), camptothecin (CPT), formaldehyde (FA), hydroxyurea (HU), or the PARP inhibitor NU-1025 was not evident (Fig S1B–F). On the other hand, PCNA[KR] cells were hypersensitive to both illudin S and irofulven (Fig 1A and B). In this study, we used both compounds,

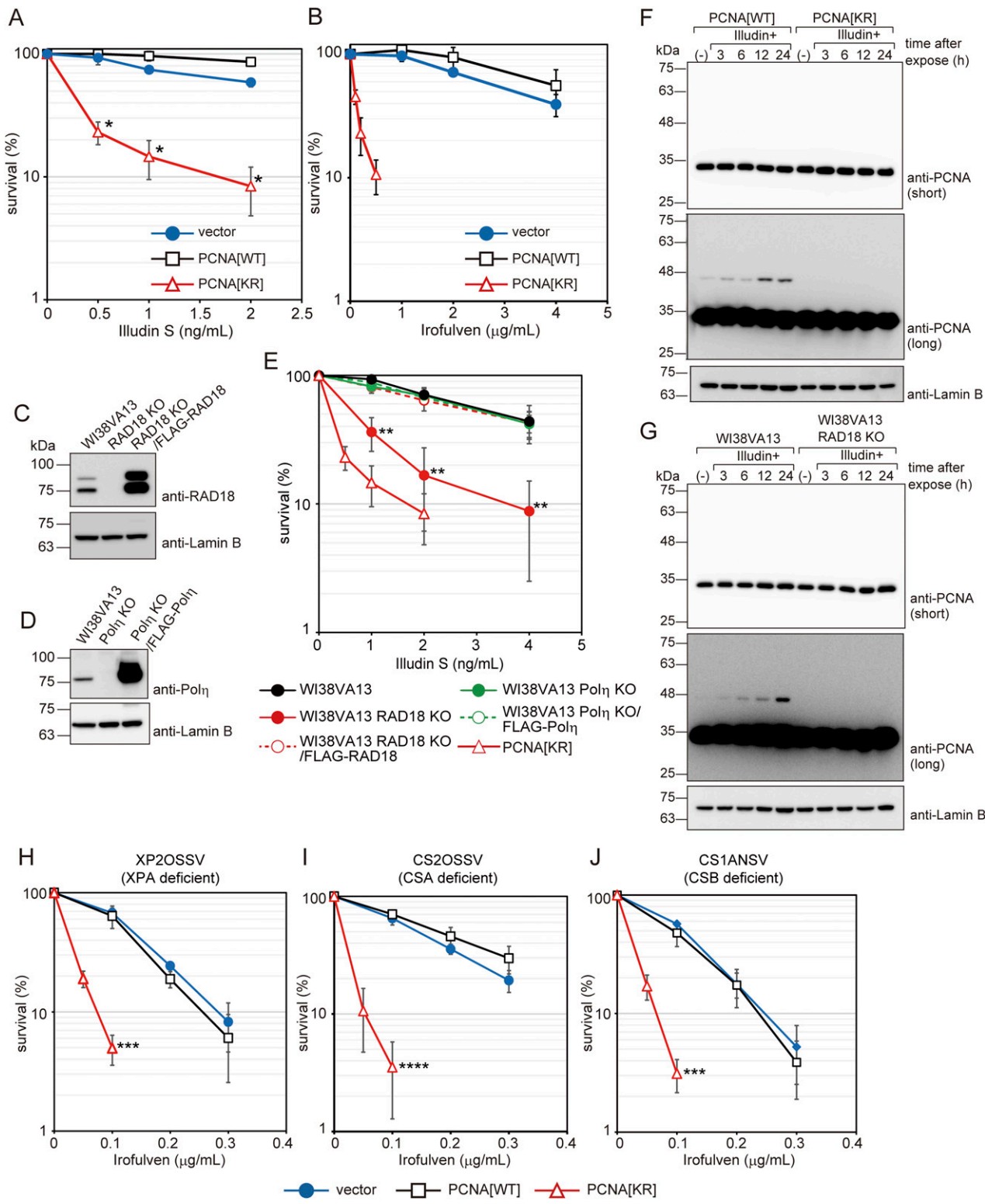

**Figure 1. PCNA ubiquitination is required for cell survival after illudin S and irofulven treatment independently of TC-NER.**
**(A, B)** WI38VA13-derived PCNA-replaced cells (PCNA[WT] or PCNA[KR]), formed by siPCNA transfection to ectopic PCNA-expressing cells, or cells harboring control empty vector (vector) were exposed to the indicated doses of illudin S for 4 d (A) or the indicated dose of irofulven for 1 h and subsequently incubated without the drug for 4 d (B). Cellular survival was evaluated by MTS assay. Data are represented as means ± SD of n = 4 (PCNA[WT] and PCNA[KR] in (A)) or n = 3 (vector in (A) and all samples in (B)) independent experiments. *P < 0.001 versus PCNA[WT]. **(C, D)** Whole-cell lysates were prepared from the cells used in (E) and analyzed by immunoblotting using anti-RAD18 and anti-Lamin B (C) or anti-Polη and anti-Lamin B1 (D) antibodies. **(E)** WI38VA13, WI38VA13 Polη KO, WI38VA13 Polη KO/FLAG-Polη, WI38VA13 RAD18 KO, and

but primarily illudin S, because of the cost of irofulven. RAD18, a major E3 ligase involved in PCNA ubiquitination, is required for resistance to the natural sesquiterpene compound illudin S, whereas Polη is not (Jaspers et al, 2002). To investigate the requirements for RAD18 and Polη in human cells, we established RAD18 KO and Polη KO WI38VA13 cells using the CRISPR-Cas9 system (Fig 1C and D). Consistent with a previous study (Jaspers et al, 2002), WI38VA13 RAD18 KO cells were more sensitive to illudin S treatment than their parental WI38VA13 cells, and resistance was restored by expression of FLAG-tagged human RAD18 (Fig 1C and E). The Polη KO cells were approximately as sensitive to illudin S as their parental human fibroblast WI38VA13 cells or FLAG-tagged human Polη-expressing Polη KO cells (Fig 1D and E). In PCNA[WT] cells, treatment with illudin S–induced PCNA mono-ubiquitination, whereas much less mono-ubiquitinated PCNA was present in PCNA[KR] cells (Fig 1F). In WI38VA13 RAD18 KO cells, the levels of mono-ubiquitinated PCNA after illudin S treatment were reduced (Fig 1G). These results suggest that illudin S induces RAD18-mediated mono-ubiquitination of PCNA at K164, which is required for cellular survival independent of Polη. The E3 ubiquitin ligases helicase-like transcription factor (HLTF) and SNF2 histone-linker PHD-finger RING-finger helicase (SHPRH) catalyze K63-linked polyubiquitination of PCNA at K164 in mammalian cells (Motegi et al, 2006, 2008; Unk et al, 2006, 2008). We detected no effect of HLTF KO, SHPRH knockdown (KD), or SHPRH KD/HLTF KO on illudin S sensitivity in human U2OS cells, whereas RAD18 KD was associated with increased illudin S sensitivity in this cell line (Fig S1G–K).

### PCNA modifications at K164 play roles distinct from those of TC-NER

Cells derived from an NER-deficient XP-A patient (XP2OSSV) and a TC-NER–deficient Cockayne syndrome group B (CS-B) patient (CS1ANSV) were sensitive to illudin S. Resistance to illudin S was restored by the expression of XPA and CSB proteins, respectively (Fig S1L–O). In contrast, cells from a GG-NER–deficient xeroderma pigmentosum group C (XP-C) patient (XP4PASV) did not exhibit elevated sensitivity to illudin S, but were highly sensitive to UV-C (Fig S1P–R). These results are consistent with a previous study showing that DNA lesions induced by illudin S or its derivatives are recognized by TC-NER but ignored by GG-NER (Jaspers et al, 2002). To explore the relationship between PCNA modifications and TC-NER, we established TC-NER–deficient cells expressing exogenous PCNA[WT] or PCNA[KR]. Replacing endogenous PCNA with exogenous PCNA[KR] in XPA-, CSA-, and CSB-deficient cells significantly increased sensitivity to irofulven (Fig 1H–J). These results indicate

that in human cells, PCNA modifications at K164 play roles outside the TC-NER pathway in response to irofulven treatment.

### PCNA modifications at K164 are required to resolve DNA replication problems induced by illudin S and irofulven

We next investigated the effect of unmodifiable PCNA on DNA replication after illudin S treatment in WI38VA13-derived cells. In these experiments, we labeled PCNA[WT] and PCNA[KR] cells in the S-phase by pulse treatment with bromodeoxyuridine (BrdU) and then followed the cell cycle progression. If the cells were not treated with illudin S, most of the labeled populations passed through the S-phase within 12 h in both PCNA[WT] and PCNA[KR] cells (Fig 2A and B). On the other hand, if the cells were treated with illudin S, S-phase progression was significantly slower in PCNA[KR] cells than in PCNA[WT] cells (Fig 2A and B). These results suggest that progression of DNA replication is severely inhibited in PCNA[KR] cells after illudin S treatment. We also examined the ability of the cells to incorporate BrdU after illudin S treatment. After treatment with illudin S for 1 h and subsequent incubation without illudin S, cells were treated with BrdU at the indicated time points (Figs 2C and S2A). In the PCNA[WT] cells, the proportion of BrdU-positive cells tended to increase after illudin S treatment, suggesting that the S-phase was somewhat delayed. Therefore, the population of S-phase cells increased and had the ability to incorporate BrdU. The BrdU intensity in the positive population was slightly reduced relative to that in untreated cells (the peak moved to the left in Fig 2C) at 3 h after illudin S treatment but was maintained until 9 h after treatment in PCNA[WT] cells. This suggests that the cells had the ability to continue DNA replication after illudin S treatment, although replication was slightly delayed. In PCNA[KR] cells, BrdU incorporation in the positive cells continued to decrease until 9 h after illudin S treatment (Figs 2C and S2A), suggesting that the PCNA[KR] cells could not overcome DNA replication problems. To further examine this phenomenon, we used a DNA fiber assay to monitor DNA synthesis after illudin S treatment. Cells were pulse labeled with chlorodeoxyuridine (CldU), treated with illudin S for 1 h, and then labeled with iododeoxyuridine (IdU) (0 h in Fig 2D). As a result, we detected only a slight reduction of IdU track length, even in PCNA[KR] cells (Fig 2D and E). DNA alkylation by illudin S and derivatives requires reduction reactions, meaning that formation of DNA adducts takes time after illudin S treatment (Pietsch et al, 2013). Therefore, we added interval periods of 1 or 3 h between illudin S treatment and IdU labeling (Fig 2D). 3 h after treatment in PCNA[KR] cells, IdU track length was significantly shorter than CldU track length; this was not observed in PCNA[WT]

WI38VA13 RAD18 KO/FLAG-RAD18 cells were exposed to illudin S for 4 d and cellular survival was evaluated by MTS assays. Data are represented as means ± SD of n = 6 (WI38VA13), n = 4 (WI38VA13 Polη KO, WI38VA13 Polη KO/FLAG-Polη, or WI38VA13 RAD18 KO), or n = 3 (WI38VA13 RAD18 KO/FLAG-RAD18) independent experiments. **P < 0.01 versus WI38VA13. **(F, G)** WI38VA13/PCNA[WT] and PCNA[KR] cells (F), and WI38VA13 and RAD18 KO cells (G) were exposed to 50 ng/ml illudin S for 1 h and incubated for the indicated periods without the drug. (−), untreated with illudin S. Whole-cell lysates were prepared and analyzed by immunoblotting using anti-PCNA and anti-Lamin B antibodies. **(H, I, J)** Empty-vector–introduced (vector) and PCNA-replaced XP2OSSV (H), CS2OSSV (I), and CS1ANSV (J) cells (PCNA[WT] or PCNA[KR]) were exposed to the indicated doses of irofulven for 1 h. After 4 d, cellular survival was evaluated by MTS assays. Data are represented as means ± SD of n = 3 independent experiments. ***P < 0.005 versus PCNA[WT]; ****P < 0.001 versus PCNA[WT]. The statistical significance was evaluated by two-tailed t tests.
Source data are available for this figure.

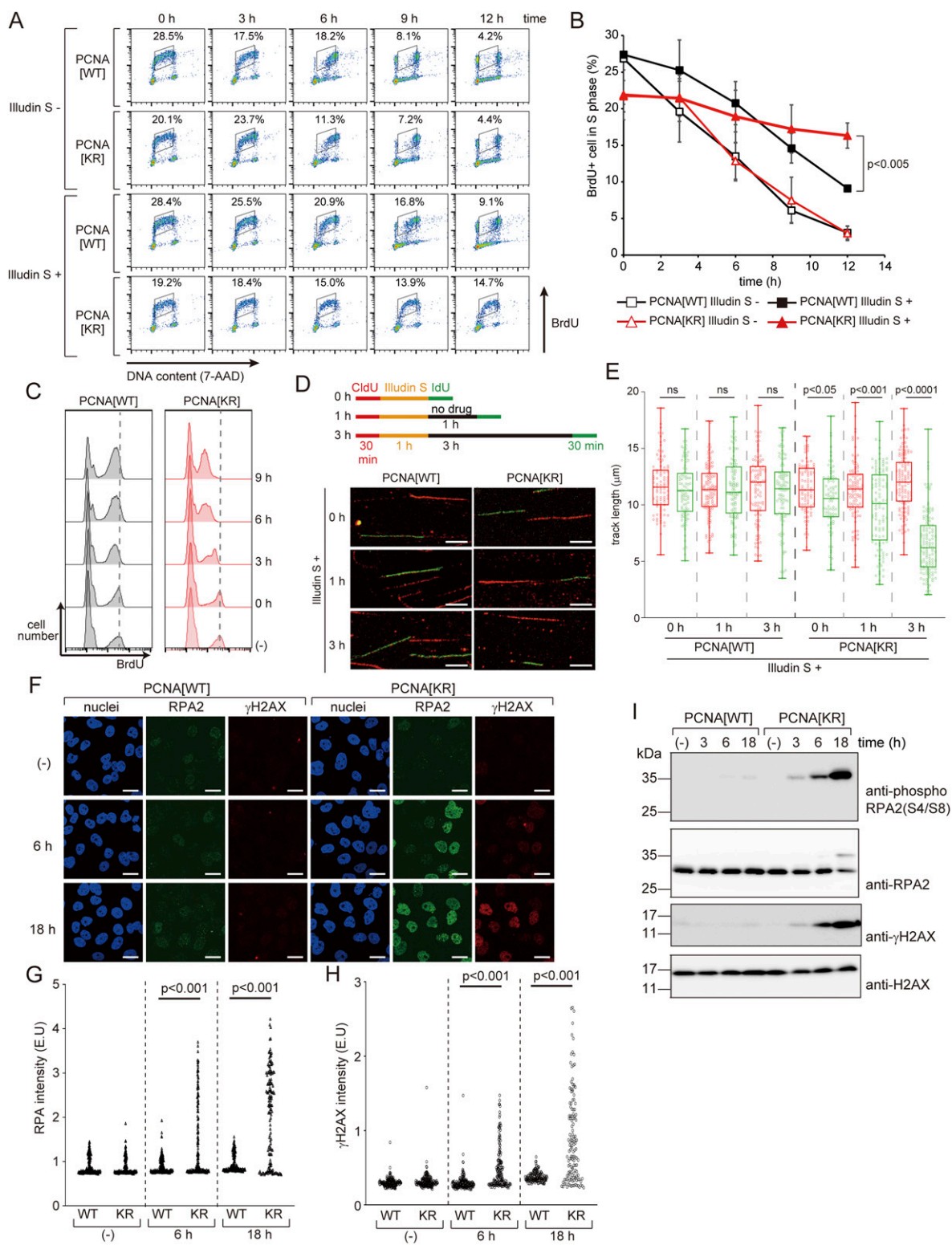

**Figure 2.  PCNA modifications are required for the progression of DNA replication after illudin S treatment.**
**(A, B)** WI38VA13/PCNA[WT] or [KR] cells were exposed to 25 ng/ml illudin S (Illudin S +) or mock medium (Illudin S −) with 20 μM BrdU for 1 h and subsequently incubated for the indicated periods without drugs. BrdU was stained with Alexa Fluor 488-conjugated anti-BrdU antibody. DNA was stained with 7-amino-actinomycin D (7-AAD). Stained cells were analyzed by fluorescence-assisted cell sorting (FACS). **(A)** FACS profiles. **(B)** The proportion of BrdU-positive S-phase cells was calculated, and data are represented as mean ± SD of n = 3 independent experiments. **(C)** WI38VA13/PCNA[WT] or PCNA[KR] cells were exposed to 25 ng/ml illudin S for 1 h and subsequently incubated for the indicated periods without the drug. Cells were treated with 20 μM BrdU for 1 h at the indicated time points, harvested, and fixed. FACS analyses were

cells (Fig 2D and E). Alteration of IdU track length was not observed without illudin S treatment (Fig S2B and C).

Blocking replicative DNA polymerases and failure to tolerate a replication block often leads to fork uncoupling and collapse, resulting in single-strand DNA gaps and double-strand breaks (Hedglin & Benkovic, 2017). To evaluate single-strand DNA accumulation, we monitored the second subunit of RPA2. Detergent-treated cells with high RPA2 signals were observed at 6 h. RPA2 signals were more abundant at 18 h after illudin S treatment in PCNA[KR] cells but not in PCNA[WT] cells (Fig 2F and G). To monitor double-strand breaks, we measured phosphorylated histone H2AX (γH2AX) by immunostaining. High γH2AX signals appeared at 6 h and were more abundant at 18 h after illudin S treatment in PCNA[KR] cells but not in PCNA[WT] cells (Fig 2F and H). Phosphorylation at Ser4/Ser8 of RPA2 and γH2AX were detected at 6 h and more abundantly at 18 h after illudin S treatment in PCNA[KR] cells (Fig 2I). These results suggest that replication blockage at lesions induced by illudin S were not overcome, and that stalled replication forks had collapsed in PCNA[KR] cells.

### Polκ and RFWD3 are involved in DNA damage tolerance

Mono-ubiquitination of PCNA controls TLS, but Polη is not required for cellular survival after illudin S treatment (Fig 1E). To determine whether other TLS polymerases are involved in bypassing illudin S–induced lesions, we tested the sensitivity of human cells in which each TLS polymerase was depleted. We found that depleting Polκ using two different siRNAs increased sensitivity to illudin S treatment, but depleting REV1, REV7, or Polι showed no significant effect (Fig 3A and B). Note that the cytotoxicity of illudin S differed between cell lines (compare Fig 3A and C). HeLaS3 Polκ KO cells were also more sensitive to illudin S treatment, and resistance was restored by ectopic expression of GFP-Polκ (Fig 3C and D). These results indicate that Polκ is required for the survival of human cells after illudin S treatment. However, the effect of Polκ depletion on illudin S resistance was smaller than that of replacing endogenous PCNA with unmodifiable PCNA (Fig 3A), prompting us to search for additional factors involved in illudin S and irofulven tolerance.

We screened a subset of a library of siRNAs targeting DNA repair factors. Human fibroblast BJ1/hTERT cells were transfected with siRNAs, and their viabilities were evaluated after irofulven treatment. To identify the factors responsible for DNA damage tolerance, we examined cellular viability under conditions of p53 or p21 co-depletion, which allows cells to enter the S phase in response to DNA damage (Cao et al, 2014). We tested two different conditions of

irofulven treatment, that is, 2 μg/ml irofulven for 1 h and 75 ng/ml irofulven for 4 d. As summarized in Table S1, REV3 and RFWD3 were listed as being high ranking in p53 or p21 co-depleted cells in both conditions. In the absence of p53 or p21 co-depletion, the effect of REV3 or RFWD3 knockdown was not apparent, suggesting that these genes were responsible for determining tolerance to irofulven-induced DNA damage. Although we confirmed that siRNA-mediated down-regulation of RFWD3 in WI38VA13 cells resulted in increased sensitivity to illudin S in this study (Fig 3E and F), we did not determine whether down-regulation of REV3 using independent siRNAs resulted in increased sensitivity to illudin S. Therefore, we focused on Polκ and RFWD3 hereafter. Consistent with a previous report in which RFWD3 KO was not established with HeLa and U2OS cells (Feeney et al, 2017), we failed to establish RFWD3 KO cells using WI38VA13 or HeLaS3 cells, suggesting that the gene is essential in these cells. Co-depletion of Polκ and RFWD3 increased sensitivity to illudin S more than single depletion in WI38VA13 (Fig 3E) and HeLa cells (Fig 5C). Either RFWD3 or Polκ depletion in TC-NER–deficient CS1ANSV cells increased illudin S sensitivity (Fig 3G), suggesting that both RFWD3 and Polκ play roles outside TC-NER.

To examine the effect of RFWD3 or Polκ depletion on DNA replication after illudin S treatment, we monitored S-phase progression in cells pulse-labeled by BrdU after illudin S treatment. Most of the labeled populations passed through the S-phase within 12 h without illudin S treatment in each siRNA-transfected cell line (Fig 3H and I). However, if the cells were treated with illudin S, S-phase progression was significantly slower in RFWD3- and Polκ-depleted cells than in cells transfected with non-targeting control siRNA (Fig 3H and I). The BrdU pulse incorporation experiment revealed that suppressing RFWD3 or Polκ decreased the ability to incorporate BrdU after treatment with illudin S (Figs 3J and S2E). RFWD3 and Polκ co-depletion tended to delay S-phase progression after illudin S treatment to a greater extent than single depletion, but the difference was not statistically significant in the flow cytometry analyses (Fig 3H and I). In the DNA fiber assay (Fig 3K), the IdU track lengths of RFWD3- or Polκ-depleted cells at 3 h after illudin S treatment were significantly reduced (P < 0.0001, versus CldU track length), whereas those of non-targeting siRNA-transfected cells were not (Fig 3K). We also observed short IdU track length using another siRNA against RFWD3 after illudin S treatment (Fig S3K and L). Alterations of IdU track length were not observed without illudin S treatment (Fig S2D). Co-depletion of RFWD3 and Polκ resulted in significantly shorter IdU track length than single depletion (P < 0.05, siRFWD3#1 versus

performed as described above. (−), untreated with illudin S. Dotted lines show the median intensity of incorporated BrdU in the untreated cells. **(D, E)** WI38VA13/PCNA [WT] or [KR] cells were labeled with 25 μM CldU for 30 min, exposed to 50 ng/ml illudin S for 1 h, incubated for 0, 1, or 3 h without the drug, and then labeled with 250 μM IdU for 30 min. Incorporated CldU and IdU were stained with anti-BrdU antibodies. **(D)** Labeling scheme of the DNA fiber assay with representative images. Scale bar, 5 μm. **(E)** Quantified CldU (red) and IdU (green) track length. At least 80 tracks from two independent experiments were evaluated. The line represents the median; boxes are the 25th and 75th percentiles; whiskers are the minimum and the maximum values. ns, not significant. **(F, G, H)** WI38VA13/PCNA[WT] or [KR] cells were exposed to 25 ng/ml illudin S for 1 h and incubated for 6 or 18 h without the drug. After eliminating the detergent-soluble fraction, the cells were fixed and stained with anti-RPA2 and anti-γH2AX antibodies. Nuclei were visualized with Hoechst 33342. **(F)** Representative images. Scale bar, 20 μm. **(G, H)** Quantified RPA2 (G) or γH2AX (H) intensities in each nucleus. At least 130 nuclei were evaluated. (−), untreated with illudin S. **(I)** WI38VA13/PCNA[WT] and PCNA[KR] cells were exposed to 25 ng/ml illudin S for 1 h and incubated for the indicated periods without the drug. (−), untreated with illudin S. Whole-cell lysates were prepared and analyzed by immunoblotting using anti-phospho RPA2 (S4/S8), anti-RPA2, anti-γH2AX, and anti-H2AX antibodies. The statistical significance was evaluated by Welch's t test (two-tailed, unpaired). Source data are available for this figure.

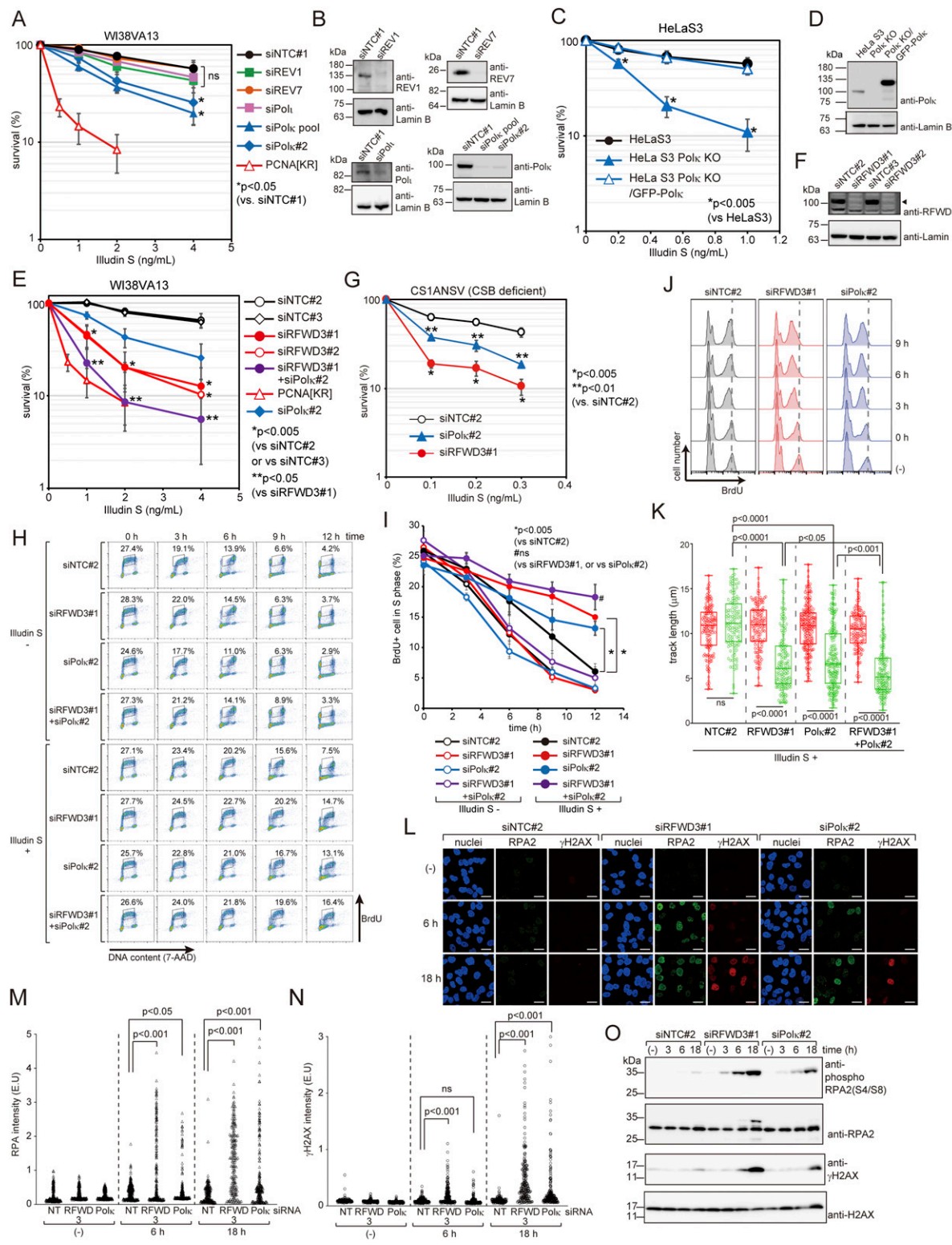

**Figure 3. Polκ and RFWD3 contribute to cellular survival and efficient DNA replication progression after illudin S and irofulven treatment.**

**(A)** WI38VA13 cells were transfected with siRNA against REV1 (siREV1), REV7 (siREV7), Polκ (siPolκ pool or siPolκ#2), Polι (siPolι), or non-targeting control siRNA (siNTC#1). Transfected cells were exposed to the indicated doses of illudin S for 4 d. Cellular survival was evaluated by MTS assay. Data are represented as means ± SD of n = 6 (siNTC#1 and siREV1), n = 4 (siPolκ pool and siPolκ#2), or n = 3 (siREV7, and siPolι) independent experiments. The data for PCNA[KR] are replotted from Fig 1C. **(B)** Whole-cell lysates were prepared from the cells used in (A) and analyzed by immunoblotting using anti-REV1, anti-REV7, anti-Polι, anti-Polκ, and anti-Lamin B antibodies. **(C)** HeLaS3, HeLaS3 Polκ KO, and HeLaS3 Polκ KO/GFP-Polκ cells were exposed to illudin S for 4 d. Cellular survival was evaluated by MTS assay. Data are represented as

siRFWD3#1+siPolκ#2; P < 0.001, siPolκ#2 versus siRFWD3#1+siPolκ#2) (Fig 3K), suggesting the independent roles of these factors. In RFWD3- or Polκ-depleted cells, we observed RPA2 accumulation in response to illudin S treatment (Fig 3L and M). We also observed RFWD3- and Polκ-depleted cells with high γH2AX signals 18 h after treatment with illudin S (Fig 3L and N). Phosphorylation of RPA2 and γH2AX were detected by immunoblotting in RFWD3- and Polκ-depleted cells after illudin S treatment (Fig 3O). These results suggest that RFWD3 and Polκ are involved in overcoming illudin S–induced DNA replication blockage.

## RFWD3 has a role independent from the FANC-pathway in DNA damage tolerance

Mutations in *RFWD3*, which was previously reported as the *FANCW* gene, result in Fanconi anemia (Knies et al, 2017). RFWD3 is involved in the FANC-mediated DNA interstrand crosslink (ICL) repair pathway by promoting HR, which is a crucial reaction in ICL repair (Feeney et al, 2017; Inano et al, 2017; Knies et al, 2017). The role of RFWD3 in ICL repair involves RPA and RAD51 ubiquitination, and requires its E3 ligase activity, chromatin localization, and/or RPA binding abilities (Inano et al, 2017). To investigate whether these activities are required for DNA damage tolerance after illudin S treatment, we established WI38VA13 cells expressing either wild-type or FLAG-RFWD3 mutants. Then, we tested their illudin S sensitivity after depleting endogenous RFWD3 with a siRNA targeting the 3′-UTR of the gene. Wild-type FLAG-RFWD3 expression rescued the illudin S sensitivity of RFWD3-depleted cells (Fig 4A and B). The C315A mutant (RFWD3[CA]) abolishes E3 ligase activity (Fu et al, 2010), whereas the I639K mutant of RFWD3 (RFWD3[IK]) has defects in chromatin localization and interaction with RPA (Inano et al, 2017; Knies et al, 2017). Neither FLAG-RFWD3[CA] nor the [IK] mutant rescued the illudin S sensitivity of RFWD3-depleted cells (Fig 4A and B), indicating that E3 ligase activity, chromatin localization, and/or RPA binding abilities of RFWD3 are required for cellular survival after illudin S treatment.

We detected ubiquitinated RPA2 after illudin S treatment, but RAD51 ubiquitination was not obvious (Fig S3A). RPA2 ubiquitination was largely dependent on RFWD3 (Fig S3A), but was independent from PCNA modification (Fig S3B). Because RPA is essential for DNA replication and cell growth (Feeney et al, 2017; Inano et al, 2017), we established WI38VA13 cells expressing wild type or mutant FLAG-RPA2 to investigate the requirement of RPA ubiquitination for DNA damage tolerance. Then, we replaced endogenous RPA2 with exogenous RPA2 using siRNA targeting the gene's 3′-UTR and tested illudin S sensitivity (Fig 4C and D). Cells in which endogenous RPA2 was replaced by wild-type FLAG-RPA2 showed similar illudin S sensitivity to control cells harboring empty vector and transfected with non-targeting siRNA (Fig 4C and D). Five lysine residues (K37/ 38/85/127/171) in RPA2 were demonstrated to be ubiquitinated and required for ICL repair (Elia et al, 2015; Inano et al, 2017). However, cells in which RPA2 was replaced by the 5KR mutant, in which the five lysine residues were mutated to arginine, showed sensitivity to illudin S that was similar to that observed in cells with wild-type RPA2 (Fig 4C and D), suggesting that RPA2 ubiquitination is not required for survival after illudin S treatment. These results imply that there are unidentified RFWD3 substrates for DNA damage tolerance that are different from those for ICL repair. Although RPA2 ubiquitination was dispensable for tolerance, cells in which RPA2 was replaced by the F248A mutant, which abrogates interactions with RFWD3 and attenuates ICL repair (Feeney et al, 2017), showed higher sensitivity to illudin S (Fig 4C and D). Together with the result that the RFWD3 I639K mutant was not able to complement illudin S sensitivity, these results indicate that the interaction between RFWD3 and RPA is required for cell survival after illudin S treatment, and is required during ICL repair (Feeney et al, 2017; Inano et al, 2017; Knies et al, 2017).

Importantly, RFWD3 depletion in FANCD2-deficient cells significantly increased illudin S sensitivity (Fig 4E), whereas the sensitivity to MMC was not increased (Fig 4F). In addition, we observed no contribution of either FANCD2 or breast cancer gene 1 (BRCA1) to survival under illudin S or irofulven treatment (Fig S3C–J). In contrast to the obvious effect of RFWD3 depletion,

---

means ± SD of n = 7 (HeLaS3) or n = 3 (HeLaS3 Polκ KO and HeLaS3 Polκ KO/GFP-Polκ) independent experiments. **(D)** Whole-cell lysates were prepared from the cells used in (C) and analyzed by immunoblotting using anti-Polκ and anti-Lamin B antibodies. **(E)** WI38VA13 cells were transfected with siRNAs against RFWD3 (siRFWD3#1 or siRFWD3#2), siRFWD3#1+siPolκ#2, or non-targeting control siRNA (siNTC#2 or siNTC#3). Transfected cells were exposed to illudin S for 4 d and cellular survival was evaluated by MTS assay. Data are represented as means ± SD of n = 7 (siRFWD3#1), n = 6 (siNTC#2), or n = 3 (siNTC#3 and siRFWD3#2) independent experiments. The data for PCNA[KR] and siPolκ#2 are replotted from Figs 1A and 3A, respectively. **(F)** Whole-cell lysates were prepared from the cells used in (E) and analyzed by immunoblotting using anti-RFWD3 and anti-Lamin B antibodies. The arrowhead shows the RFWD3 signal. **(G)** CS1ANSV cells were transfected with siNTC#2, siRFWD3#1, or siPolκ#2. Transfected cells were exposed to illudin S for 4 d and cellular survival was evaluated by MTS assay. Data are represented as means ± SD of n = 3 independent experiments. **(H, I)** WI38VA13 cells were transfected with siRFWD3#1, Polκ#2, siRFWD3#1+siPolκ#2, or NTC#2. Cells were exposed to 25 ng/ml illudin S and 20 μM BrdU for 1 h and incubated for indicated periods without the drugs. Cells were analyzed as described in Fig 2A and B. **(H)** FACS profiles. **(I)** The proportion of BrdU-positive S-phase cells was calculated. Data are represented as mean ± SD of n = 3 independent experiments. **(J)** WI38VA13 cells were transfected with siRFWD3#1, siPolκ#2, or siNTC#2. Cells were exposed to 25 ng/ml illudin S for 1 h and then incubated for the indicated periods without the drug. Cells were treated with 20 μM BrdU for 1 h at the indicated time points, harvested, and fixed. FACS analyses were performed as described in Fig 2C. BrdU intensities are shown. (−), untreated sample. Dotted lines show the median intensity of incorporated BrdU in untreated cells. **(K)** WI38VA13 cells were transfected with siRFWD3#1, Polκ#2, siRFWD3#1+siPolκ#2, or NTC#2. Cells were labeled with 25 μM CldU for 30 min, exposed to 50 ng/ml illudin S for 1 h, incubated for 3 h without the drug, and labeled with 250 μM IdU for 30 min. Incorporated CldU and IdU were stained with anti-BrdU antibodies. Quantified CldU (red) and IdU (green) track length were shown. At least 100 tracks from two independent experiments were evaluated. The line represents the median; boxes are the 25th and 75th percentiles; whiskers are the minimum and the maximum values. **(L, M, N)** siRFWD3#1, siPolκ#2, or siNTC#2-transfected WI38VA13 cells were exposed to 25 ng/ml illudin S for 1 h and incubated for 6 or 18 h without the drug. RPA and γH2AX were detected and quantified as described in Fig 3D–F. **(L)** Representative images. Scale bar represents 20 μm. **(M, N)** Quantified RPA (M) or γH2AX (N) intensities in each nucleus. At least 150 nuclei were evaluated. **(O)** WI38VA13 cells were transfected with siRFWD3#1, siPolκ#2, or siNTC#2, exposed to 25 ng/ml illudin S for 1 h, and incubated for the indicated periods without the drug. (−), untreated with illudin S. Whole-cell lysates prepared and analyzed by immunoblotting using anti-phospho RPA2 (S4/S8), anti-RPA2, anti-γH2AX, and anti-H2AX antibodies. The statistical significance was evaluated by two-tailed t tests. ns, not significant. Source data are available for this figure.

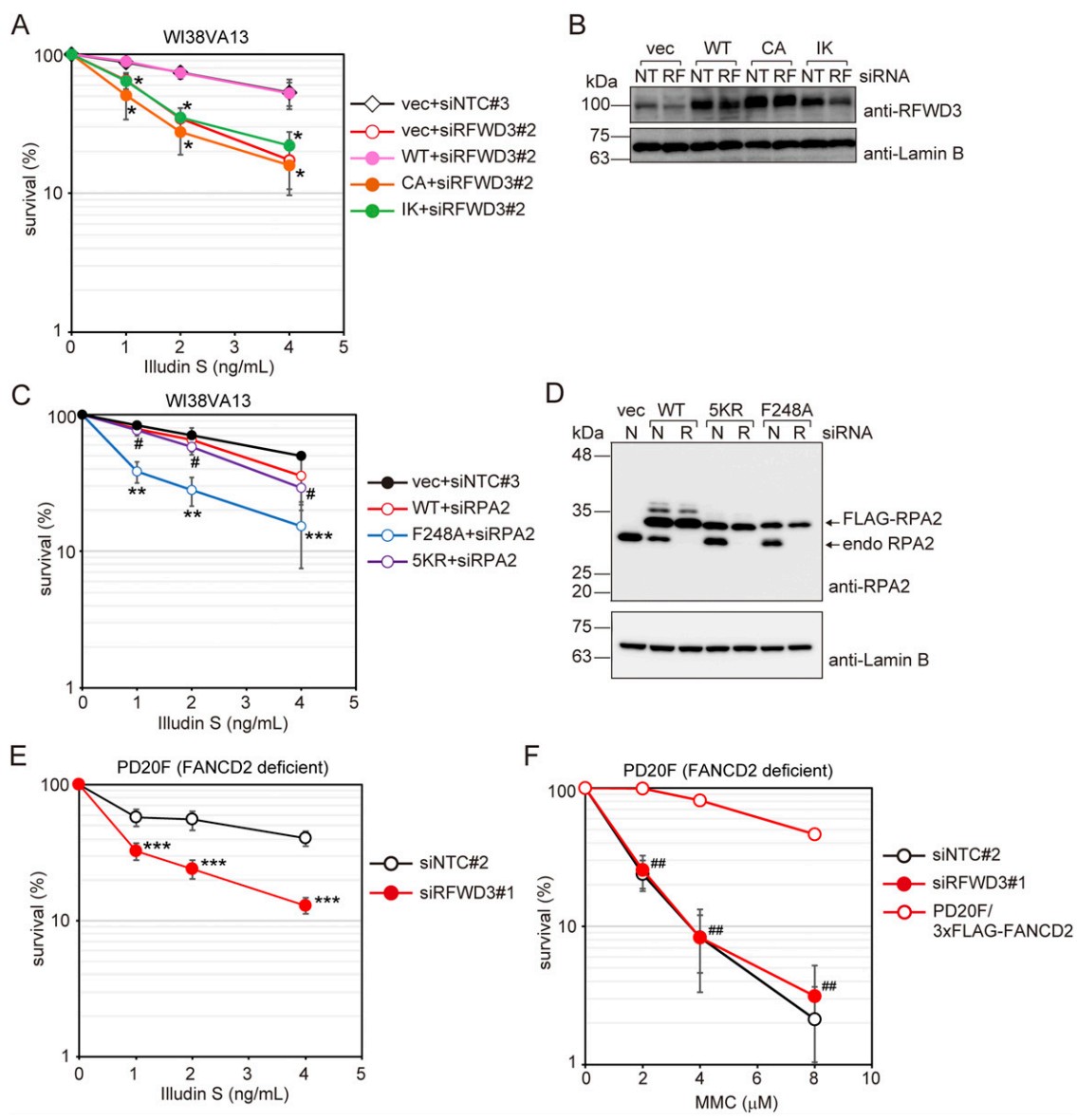

**Figure 4. RFWD3 contributes to cellular survival after illudin S treatment independently of the FANC-pathway.**
**(A)** WI38VA13 cells stably expressing wild-type (WT) or mutant FLAG-RFWD3 were transfected with siRNA targeting the 3′-UTR of RFWD3 (siRFWD3#2). Cells harboring empty vector (vec) were transfected with siRFWD3#2 or non-targeting control siRNA (siNTC#3). Transfected cells were exposed to illudin S for 4 d. Cellular survival was evaluated by MTS assay. Data are represented as means ± SD of n = 4 independent experiments. CA: C315A; IK: I639K. *$P < 0.01$ versus WT+siRFWD3#2. **(B)** WI38VA13 cells stably expressing wild-type (WT), mutant FLAG-RFWD3, or harboring the empty vector were transfected with siRFWD3#2 or siNTC#3. Whole-cell lysates were prepared and analyzed by immunoblotting using anti-RFWD3 and anti-Lamin B antibodies. NT: siNTC#3, RF: siRFWD3#2. **(C)** WI38VA13 cells stably expressing wild-type (WT) or mutant FLAG-RPA2 were transfected with siRNA targeting the 3′-UTR of RPA2 (siRPA2). Cells harboring empty vector (vec) were transfected with non-targeting control siRNA (siNTC#3). Transfected cells were exposed to illudin S for 4 d. Cellular survival was evaluated by MTS assay. Data are represented as means ± SD of n = 3 (vec) or n = 4 (others) independent experiments. 5KR: K37R/K38R/K85R/K127R/K171R. **$P < 0.01$ versus WT+siRPA2; ***$P = 0.09$ versus WT+siRPA2; #, not significant versus WT+siRPA2. **(D)** WI38VA13 cells stably expressing wild-type (WT), mutant FLAG-RPA2, or harboring the empty vector were transfected with siRPA2 or siNTC#3. Whole-cell lysates were prepared and analyzed by immunoblotting using anti-RPA2 and anti-Lamin B antibodies. N: siNTC#3, R: siRPA2. **(E, F)** PD20F cells were transfected with siRFWD3#1 or siNTC#2. Transfected cells were exposed to the indicated doses of illudin S for 4 d (E) or the indicated doses of MMC for 1 h and subsequently incubated without the drug for 4 d (F). Cellular survival was evaluated by MTS assay. Data are represented as means ± SD of n = 3 independent experiments. ***$P < 0.05$ versus siNTC#2; ##, not significant versus siNTC#2. Statistical significance was evaluated by two-tailed $t$ test.
Source data are available for this figure.

BRCA1 or RAD51 depletion did not affect IdU track length after illudin S treatment in WI38VA13 cells (Fig S3K and L). These results strongly suggest that RFWD3 acts in the same pathway as FANCD2 in ICL repair but plays a FANCD2- and canonical HR-independent role in tolerance to illudin S–induced DNA lesions.

### RFWD3- and Polκ-mediated DNA damage tolerance pathways require PCNA modifications at K164

To investigate the relationship among PCNA modifications at K164, RFWD3, and Polκ in cellular DNA damage tolerance, we tested the illudin S sensitivities of RFWD3- or Polκ-depleted PCNA[KR] cells (Fig 5A and B). RFWD3 or Polκ knockdown in PCNA[WT] cells increased sensitivity to illudin S. In contrast, the effects of RFWD3 or Polκ knockdown on illudin S sensitivity were not significant in PCNA [KR] cells. These observations suggest that the function of RFWD3 and Polκ depend on PCNA modifications. We then depleted RFWD3 in HeLaS3 Polκ KO cells and parental HeLaS3 cells (Fig 5C). In both cell types, RFWD3 depletion increased sensitivity to illudin S, suggesting that RFWD3 has a role outside of Polκ in tolerance to illudin S damage. Co-depletion of RFWD3 and Polκ altered illudin S sensitivity to approximately the same level as that in PCNA[KR] cells (Fig 5C), suggesting that RFWD3- and Polκ-mediated pathways are almost solely responsible for PCNA modification–dependent tolerance of DNA damage caused by illudin S treatment.

Gallina et al (2021) showed that RFWD3 contributes to PCNA ubiquitination in vitro and in human U2OS cells (Gallina et al, 2021). To address the possibility that PCNA is a substrate of RFWD3 after illudin S treatment, we monitored PCNA mono- (Fig S4) and poly-ubiquitination (Fig S5) in WI38VA13, U2OS, and HeLaS3 cells. Mono-ubiquitinated PCNA levels were higher in RFWD3-depleted WI38VA13 and HeLaS3 cells than in non-targeting cells after illudin S treatment, but not in U2OS cells (Fig S4A–F). The formation of DNA adducts depends on cellular ability to reduce illudin S (Gong et al, 2007), meaning the level of DNA damage may differ between cells. In addition, increased DNA replication blockage in RFWD3-depleted cells after illudin S treatment can result in increasing PCNA mono-ubiquitination. Therefore, we could not determine whether RFWD3 directly contributes to PCNA mono-ubiquitination. Under UV irradiation, which produces equivalent amounts of DNA lesions in each cultured cell line, RFWD3 depletion did not significantly affect mono-ubiquitinated PCNA levels in WI38VA13, U2OS, or HeLa cells (Fig S4G–L). No obvious differences were seen in the protein levels of PCNA, RAD18, Polκ, or Polη after illudin S treatment or UV irradiation (Fig S4A–L). To evaluate PCNA poly-ubiquitination, we performed cell fractionation after illudin S treatment or UV irradiation and analyzed the chromatin fraction (Fig S5). In U2OS cells, poly-ubiquitinated PCNA after illudin S treatment was slightly reduced by siRFWD3 transfection, although this effect was not evident in WI38VA13 cells (Fig S5A and C). After UV irradiation, poly-ubiquitinated PCNA was slightly reduced by siRFWD3#1 transfection in both cell lines, although the effects of siRFWD3#2 transfection were lesser (Fig S5B and D). PCNA poly-ubiquitination level was not altered in SHPRH-depleted HLTF KO cells after illudin S treatment (Fig S5E). These findings suggest that RFWD3 contributed to PCNA poly-ubiquitination after illudin S treatment.

### RFWD3 plays a role in tolerance to UV-induced DNA damage outside of Polη

To clarify whether the role of RFWD3 in DNA damage tolerance is specific for illudin S and its derivatives, we examined UV sensitivity in RFWD3-depleted cells. As shown in Fig 6A, RFWD3 depletion increased cellular sensitivity after UV irradiation. Polη KO cells showed increased sensitivity to UV irradiation compared with parental WI38VA13 cells. This sensitivity was rescued by ectopic expression of FLAG-Polη (Fig 6A). RFWD3 depletion also increased UV sensitivity in Polη KO cells, suggesting that RFWD3 has a role distinct from Polη during cellular survival after UV irradiation. We investigated the effect of RFWD3 depletion on the S-phase progression after UV irradiation. No significant differences in the S-phase progression of the BrdU-labeled population were observed in cells without UV irradiation between non-targeting control siRNA-transfected cells and RFWD3- or Polη-depleted cells (Fig 6B and C). As expected, the progression of BrdU-labeled Polη knockdown cells was slower than that of control cells after UV irradiation (Fig 6B and C). We also observed that RFWD3 depletion slowed the progression of BrdU-labeled cells after UV irradiation, suggesting that RFWD3 is involved in tolerance to UV-induced DNA damage, as observed in illudin S–treated cells. The S phase progression of RFWD3 and Polη co-depleted cells was not statistically different from that in single knockdown cells after UV irradiation, according to flow cytometry analyses (Fig 6B and C). We then tested DNA synthesis after UV irradiation by the DNA fiber assay. Cells were treated with CldU and irradiated with UV after IdU treatment (Fig 6D). Without UV irradiation, no significant differences in the IdU/CldU ratio were observed between control siRNA-transfected cells and RFWD3- or Polη-depleted cells (Fig 6D and E). After UV irradiation, the IdU/CldU ratio decreased in Polη-depleted cells compared with that in control cells (Fig 6D and E), suggesting that DNA synthesis was blocked in Polη-depleted cells after UV irradiation. The IdU/CldU ratio of RFWD3-depleted cells also decreased significantly after UV irradiation compared with control cells (P < 0.0001) (Fig 6D and E). Importantly, RFWD3 and Polη co-depletion significantly decreased the IdU/CldU ratio compared with single depletion (P < 0.0001) (Fig 6D and E). Although RFWD3 knockdown in PCNA[WT] cells increased UV sensitivity, there was no significant effect in PCNA[KR] cells (Fig 6F), suggesting that RFWD3 plays a role in tolerance to UV-induced DNA damage that depends on PCNA modifications. These results suggest that RFWD3- and Polη-mediated DNA damage tolerance pathways to UV-induced DNA lesions are independent of each other, but both depend on PCNA modifications at K164.

Our results suggest that PCNA modifications at K164 generally contribute to DNA damage tolerance involving RFWD3 and TLS polymerases appropriate for the type of DNA lesion (Fig 6G).

## Discussion

### Detection of PCNA modification–dependent, Polη-independent DNA damage tolerance pathways using illudin S and irofulven

In human cells, PCNA modifications, mainly mono-ubiquitination, are required for DNA damage tolerance from UV-irradiation. Polη plays a major role in bypassing replication blockage by UV-induced lesions (Kanao et al, 2015a). To investigate the PCNA modification–dependent, Polη-independent DNA damage tolerance pathways, we searched for a form of DNA damage in which cells require PCNA ubiquitination but not Polη to survive. We found that PCNA

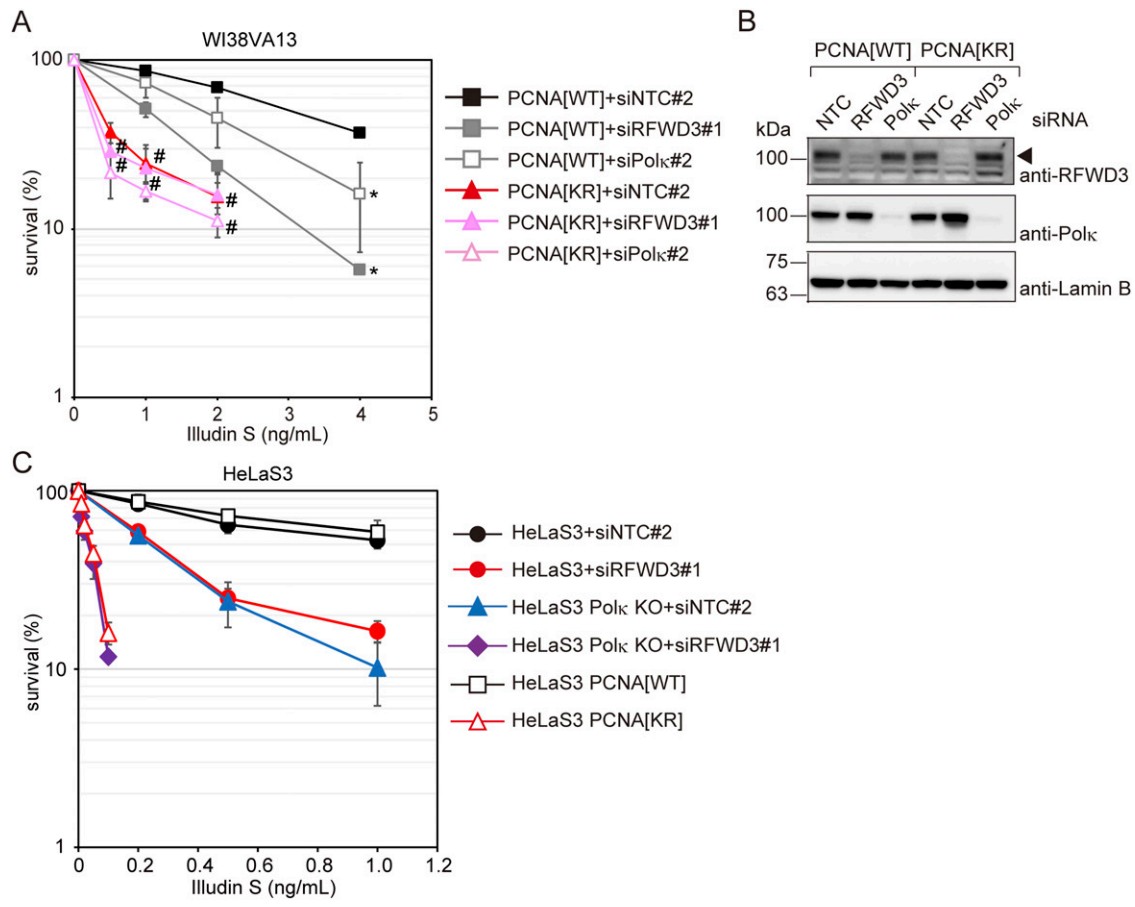

**Figure 5.    RFWD3- and Polκ-mediated DNA damage tolerance pathways are independent of each other, but both are dependent on PCNA modifications at K164.**
**(A)** PCNA[WT] and [KR] cells were transfected with siRFWD3#1, siPolκ#2, or siNTC#2. Cells were exposed to illudin S for 4 d and analyzed by MTS assay. Data are represented as means ± SD of n = 4 independent experiments. *P < 0.05 versus siNTC#2; #, not significant versus siNTC#2. **(B)** Whole-cell lysates from the cells using (A) were prepared and analyzed by immunoblotting using anti-RFWD3, anti-Polκ, and anti-Lamin B1 antibodies. The arrowhead shows the RFWD3 signal. **(C)** HeLaS3 and HeLaS3 Polκ KO cells were transfected with siRFWD3#1 or siNTC#2 and exposed to illudin S for 4 d. HeLaS3 PCNA[WT] or PCNA[KR] cells were exposed to illudin S for 4 d. Cellular survival was evaluated by MTS assay. Data are represented as means ± SD of n = 4 (HeLaS3+siNTC#2, HeLaS3+siRFWD3#2, and HeLaS3 Polκ KO+siNTC#2) or n = 3 (HeLaS3 Polκ KO+siRFWD3#1, HeLaS3 PCNA[WT], and PCNA[KR]) independent experiments. Statistical significance was evaluated by two-tailed *t* test.
Source data are available for this figure.

modification–deficient cells were hypersensitive to illudin S and its derivative, irofulven. These compounds induce bulky alkyl DNA adducts. A previous study showed that XP-V fibroblasts are not hypersensitive to illudin S, whereas RAD18 deficiency in chicken DT40 cells increases illudin S sensitivity (Jaspers et al, 2002). Consistently, RAD18 KO increased cellular sensitivity to illudin S in human cells, whereas Polη KO did not, suggesting that PCNA ubiquitination-dependent, Polη-independent DNA damage tolerance pathways are involved in protecting cells against these compounds.

Illudin S and its derivatives block RNA synthesis, but MMS, a widely used DNA alkylator, does not (Malvezzi et al, 2017b). One possible explanation is that illudin S and its derivatives produce relatively large alkyl adducts, whereas MMS causes DNA methylation. Indeed, an in vitro study showed that purified yeast Pol II stalled at 3d-Napht-A, a model adduct of illudin S, in template DNA but not at 3-deaza-3-methyl-adenosine (3d-Me-A), a model of 3-methyladenine, which is a major alkylated DNA adduct created by MMS (Malvezzi et al, 2017b). Defects in TC-NER result in cellular

hypersensitivity to illudin S or its derivatives (Jaspers et al, 2002; Koeppel et al, 2004; Schwertman et al, 2012). Indeed, these compounds produce DNA lesions that inhibit transcription and repair by TC-NER. In contrast, these lesions are ignored by GG-NER, meaning that they remain in the global genome and could block DNA replication. Consistent with this idea, illudin S impedes DNA replication in human cells (Kelner et al, 1987). In light of our observation that PCNA modifications are required for the progression of replication after treatment with illudin S and that PCNA modification plays roles outside TC-NER (Fig 6G), we conclude that illudin S and irofulven are useful for studying unidentified DNA damage tolerance pathways that are mediated by PCNA modifications.

### Involvement of Polκ in PCNA modification–dependent DNA damage tolerance

Y-family polymerases preferentially interact with mono-ubiquitinated PCNA (Kanao & Masutani, 2017). We observed that

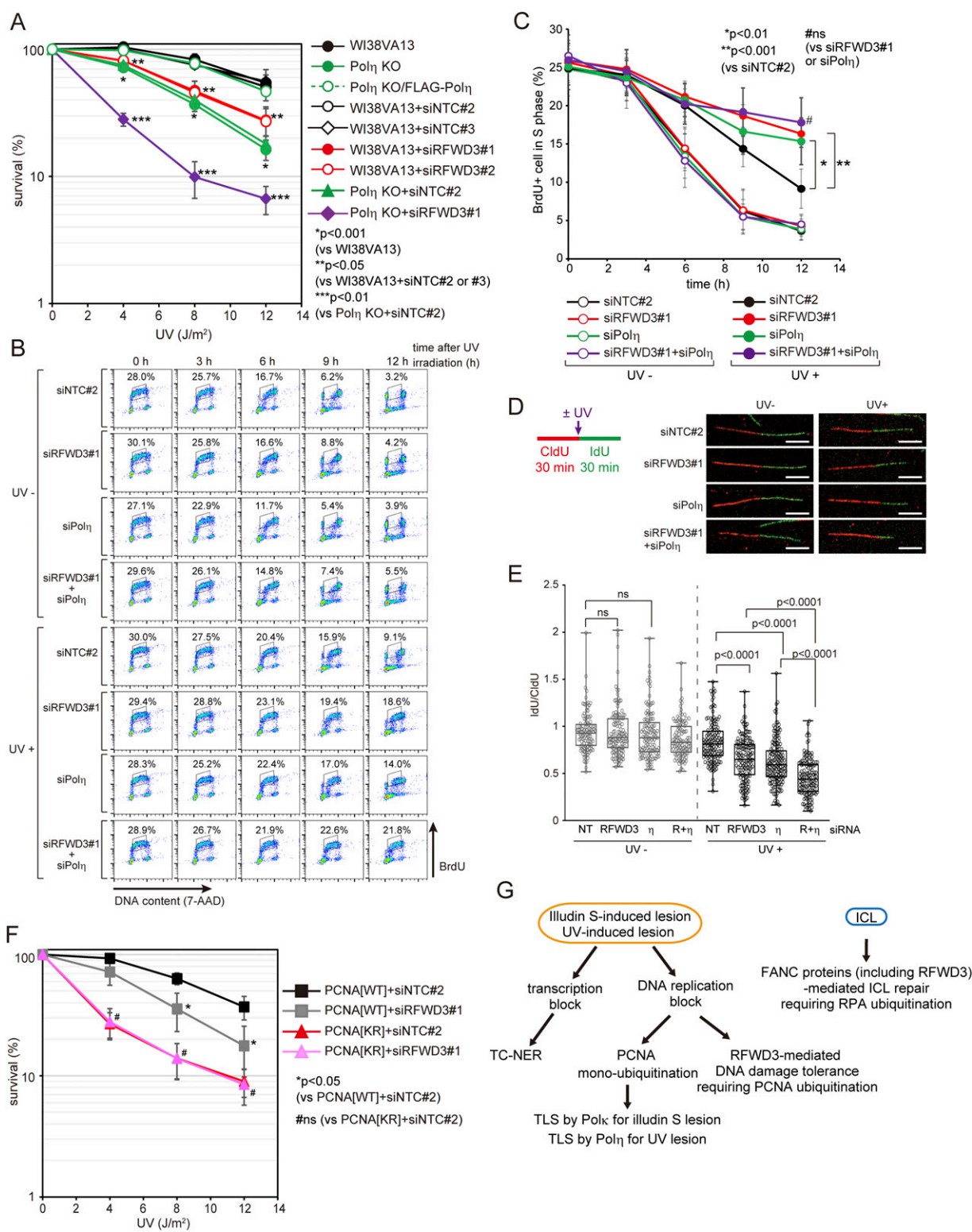

**Figure 6. RFWD3 participates in DNA damage tolerance of UV-induced DNA damage dependent on PCNA modifications at K164.**

**(A)** WI38VA13 cells were transfected with siNTC#2, siNTC#3, siRFWD3#1, or siRFWD3#2. WI38VA13 Polη KO cells were transfected with siNTC#2 or siRFWD3#1. The transfected cells and WI38VA13, WI38VA13 Polη KO, or WI38VA13 Polη KO/FLAG-Polη cells were irradiated with the indicated dose of UV-C and subsequently incubated for 4 d. Cellular survival was evaluated by MTS assay. Data are represented as means ± SD of n = 5 (WI38VA13), n = 4 (Polη KO, Polη KO/FLAG-Polη, and WI38VA13+siNTC#3), or n = 3 (WI38VA13+siNTC#2, WI38VA13+siRFWD3#1, WI38VA13+siRFWD3#2, Polη KO+siNTC#2, and Polη KO+siRFWD3#1) independent experiments. **(B, C)** WI38VA13 cells were transfected with siRFWD3#1, siPolη, siRFWD3#1+siPolη, or siNTC#2. Cells were irradiated with 4 J/m² UV light, exposed to 20 μM BrdU for 1 h, and incubated for the indicated

Polκ is involved in overcoming replication blockage caused by illudin S–induced DNA damage and is dependent on PCNA modification. However, we did not detect that Polη, Polι, REV1, or REV7, a subunit of Polζ, contribute to cellular survival after illudin S treatment (Figs 1E and 3A). Olivieri et al reported REV3, RFWD3, RAD18, REV7, REV1, and Polκ as genes whose loss caused sensitivity to illudin S treatment (Olivieri et al, 2020). The discrepancy regarding REV1 or REV7 may be explained by incomplete depletion via siRNA in our experiments. REV3 was listed in our first screening (Table S1), but we failed to confirm its role with independent siRNAs. Therefore, we do not exclude the possibility that Polζ and REV1 are involved in DNA damage tolerance under illudin S and/or irofulven treatment.

In vitro, Polκ can incorporate a nucleotide opposite 3d-Napht-A to bypass an existing lesion (Malvezzi et al, 2017a), suggesting that Polκ is involved in bypassing illudin S– or irofulven-induced lesions in cells. Polκ can bypass the benzo[*a*]pyrene-induced DNA adduct, thus protecting cells from benzo[*a*]pyrene exposure (Zhang et al, 2000; Ogi et al, 2002; Rechkoblit et al, 2002). Structural analysis revealed that Polκ easily accommodates the benzo[*a*]pyrene adduct in its active site without steric hindrance (Jha et al, 2016). Based on the structure of Polκ bound to benzo[*a*]pyrene-dG, Malvezzi et al (2017a) computationally modeled the structure of Polκ and 3d-Napht-A and showed that 3d-Napht-A is accommodated in the active site of Polκ without steric clash (Malvezzi et al, 2017a). Polη and Polι can incorporate nucleotides opposite 3d-Napht-A, but steady-state kinetics revealed that the efficiency of the correct dTTP insertion opposite 3d-Napht-A is 38-fold higher for Polκ ($k_{cat}/K_m$ = 0.8) than for Polη ($k_{cat}/K_m$ = 2.1 × $10^{-2}$). This result suggests a predominant role of Polκ in bypassing DNA lesions induced by illudin S and irofulven (Malvezzi et al, 2017a). Polζ cannot incorporate a nucleotide opposite 3d-Napht-A (Malvezzi et al, 2017a); however, the ability of Polζ to act as an extender for that adduct has not been examined. Together with these observations, our results suggest that, in human cells, Polκ is responsible for TLS of illudin S– and irofulven-induced bulky DNA adducts, and that this process is dependent on PCNA modification (Fig 6G).

### RFWD3 activity in PCNA modification–dependent DNA damage tolerance is distinct from the FANC pathway

The effect of Polκ depletion was smaller than the effect of defects in PCNA modification, suggesting that TLS by Polκ is not the sole pathway for overcoming replication blockage by these lesions. We found that RFWD3 participates in a branch of PCNA modification–dependent DNA damage tolerance in human cells. RFWD3 is the product of a gene that is mutated in Fanconi anemia

syndrome (Knies et al, 2017) and plays a crucial role in HR during stalled replication fork repair and ICL repair (Elia et al, 2015; Inano et al, 2017). Notably, deficiency of FANCD2 or BRCA1, both of which are essential for ICL repair, increases cellular sensitivity to irofulven, although the differences between the sensitivities of deficient and complemented cells are not large (Wang et al, 2006; Wiltshire et al, 2007). In this study, we demonstrate that BRCA1 and FANCD2 make smaller contributions than RFWD3 to cellular survival after illudin S or irofulven treatment (Fig S3). More importantly, our experiments clearly demonstrate that RFWD3 makes a significant contribution to tolerance, even in FANCD2-deficient cells (Fig 4E), indicating that RFWD3 plays a role in DNA damage tolerance outside the FANC pathway (Fig 6G). Although E3 ligase activity and interaction with RPA and/or chromatin localization are common features of RFWD3 function in illudin S tolerance (Fig 4A) and ICL repair (Feeney et al, 2017; Inano et al, 2017), RPA ubiquitination is dispensable for tolerance (Fig 4C) and required for ICL repair (Inano et al, 2017). These findings strongly suggest that different sets of proteins are ubiquitinated by RFWD3 during ICL repair and tolerance. The crucial targets of RFWD3-mediated ubiquitination in tolerance need to be elucidated to understand the precise mechanism.

Gallina et al (2021) showed that RFWD3 is required for error-prone TLS across DNA–protein crosslinks, ICL, and CPD in an in vitro system using *Xenopus* egg extract (Gallina et al, 2021). This study demonstrated that RFWD3 stimulates PCNA poly-ubiquitination and facilitates TLS polymerase accumulation to damage sites, although RFWD3 is not responsible for initial PCNA mono-ubiquitination (Gallina et al, 2021). Considering our results and those of Gallina et al, it is unlikely that RFWD3 is the E3 ligase responsible for PCNA mono-ubiquitination; however, it may be involved in PCNA poly-ubiquitination after illudin S treatment. How PCNA ubiquitination and RFWD3 are coordinated during DNA damage tolerance remains an important issue to be addressed in the future.

### General future of PCNA modification–dependent DNA damage tolerance

In this study, we showed that illudin S and irofulven are useful resources for studying PCNA modification–mediated mechanisms for overcoming replication blockage. Various DNA lesions on the template DNA strand block the progression of DNA polymerases. TLS by Polη is directed by mono-ubiquitinated PCNA and is largely responsible for tolerance to CPDs, which are the major DNA lesions induced by UV-irradiation. However, Polη-mediated TLS is dispensable for lesions induced by illudin S and irofulven. Instead, for those lesions, Polκ-mediated TLS and DNA damage tolerance

---

periods. The cells were analyzed as described in Fig 3A and B. **(B)** FACS profiles. **(C)** The proportion of BrdU-positive S-phase cells was calculated. Data are represented as means ± SD of n = 3 independent experiments. **(D, E)** WI38VA13 cells were transfected with siRFWD3#1, siPolη, siRFWD3#1+siPolη, or siNTC#2. Cells were labeled with 25 μM CldU for 30 min, irradiated with 8 J/m$^2$ UV light, and labeled with 250 μM IdU for 30 min. Incorporated CldU and IdU were stained with anti-BrdU antibodies. **(D)** Labeling scheme of DNA fiber assay and representative images. Scale bar, 5 μm. **(E)** The ratio between CldU and IdU track length are shown. At least 100 tracks from two independent experiments were evaluated. The line represents the median; boxes are the 25th and 75th percentiles; whiskers are the minimum and the maximum. η; Polη, R+η; RFWD3+Polη. **(F)** PCNA[WT] and [KR] cells were transfected with siRFWD3#1 or siNTC#2. The cells were irradiated with the indicated dose of UV-C and subsequently incubated for 4 d. Cellular survival was evaluated by MTS assay. Data are represented as the mean ± SD of n = 4 (PCNA[WT]+siRFWD3#1 and PCNA[KR]+siRFWD3#1) or n = 3 (PCNA[WT]+siNTC#2 and PCNA[KR]+siNTC#2) independent experiments. **(G)** Model of DNA damage tolerance pathways for illudin S–induced, UV-induced, and ICL DNA lesions. Blockage of DNA replicative polymerases is resolved by two pathways for DNA damage tolerance. One pathway involves TLS, in which lesion-specific DNA polymerases, Polη for UV- and Polκ for illudin S lesions, respectively, resolve the blocked DNA replication. The other pathway is mediated by RFWD3 and is distinct from FANC pathway. Statistical significance was evaluated by two-tailed *t* test. ns, not significant.

involving RFWD3 are promoted by PCNA modifications. Importantly, and in line with the previous study (Gallina et al, 2021), RFWD3 also contributed to tolerance after UV-irradiation, independent of Polη. Our observations suggest that cells may have two general pathways underlying the tolerance to DNA replication blockage caused by various lesion types. One pathway is mediated by TLS polymerases appropriate to the lesion type, and the other is the RFWD3-mediated pathway. Both pathways require PCNA modifications (Fig 6G). However, it should be noted that this study was limited in that we could examine the RFWD3 requirements only through use of siRNAs because of the difficulty to establish *RFWD3* KO in human cell lines (Feeney et al, 2017; Knies et al, 2017). Therefore, we do not exclude the possibility that RFWD3 plays a role upstream of DNA damage tolerance branches by controlling PCNA ubiquitination.

Because a point mutation in RFWD3 at the residue crucial for its function in ICL repair causes Fanconi anemia (Knies et al, 2017), it is possible that mutations in RFWD3 at sites associated with DNA damage tolerance cause photosensitive disorders.

# Materials and Methods

### Plasmids

The FLAG-Polη and GFP-Polκ expression constructs pIRESneo2-FLAG-Polη and pAcGFP/Polκ, respectively, were prepared as previously described (Masuda et al, 2015). The siRNA-resistant PCNA expression constructs pMK10/PCNA and pMK10/PCNA[K164R] were prepared as previously described (Kanao et al, 2015a). Human *RAD18* cDNA was cloned into pIREShyg3 (Takara Bio) to obtain pIREShyg3/FLAG-RAD18. To obtain pX459/RAD18, pX459/Polη, and pX459/Polκ for CRISPR-Cas9 modification, annealed oligonucleotides (RAD18: 5′-CACCATAGATGATTTGCTGCGGTG-3′ [forward], 5′-AAACCACCG-CAGCAAATCATCTATC-3′ [reverse]; Polη: 5′-CACCGCACAAGTTCGTGAGTCCCG-3′ [forward], 5′-AAACCGGGACTCACGAACTTGTGC-3′ [reverse]; Polκ: 5′-CACCGAGGGACAATCCAGAATTGA-3′ [forward], 5′-AAACTCAATTCTG-GATTGTCCCTC-3′ [reverse]) were cloned into pSpCas9(BB)-2A-Puro (pX459) V2.0 (Ran et al, 2013) (Addgene). The annealed oligonucle-otides (5′-CTAGCCATATGGACTACAAAGACGATGACGACAAGG-3′ [for-ward], 5′-AATTCCTTGTCGTCATCGTCTTTGTAGTCCATATGG-3′ [reverse]) were cloned into pIRESneo2 (Takara Bio) to obtain pIRESneo2/FLAG, into which cDNA for human *RFWD3* (Promega) was cloned to obtain pIRESneo2/FLAG-RFWD3. Constructs for expression of mutant RFWD3 were generated using the following primers: C315A: 5′-CTTTGGGTA-TAGGGCCATTTCCACG-3′ (forward), 5′-CGTGGAAATGGCCCTATACCCAAAG-3′ (reverse); I639K: 5′-AGGGGGCTGCAAAGACTTTCAG-3′ (forward), 5′-CTGAAAGTCTTTGCAGCCCCCT-3′ (reverse). Human *RPA2* cDNA was cloned into pIRESneo2/FLAG to obtain pIRESneo2/FLAG-RPA2. Constructs for RP2 mutants were generated using the following primers: K37R/K38R: 5′-CTCAAGCCGAAAGGAGATCTAGAGCCCGAGC-3′ (forward), 5′-GCTCGGGCTCTAGATCTCCTTTCGGCTTGAG-3′ (re-verse); K85R: 5′-CAGACATGCAGAGCGGGCCCCAACCAACATTG-3′ (forward), 5′-CAATGTTGGTTGGGGCCCGCTCTGCATGTCTG-3′ (reverse); K127R: 5′-CCTCCAGAAACATACGTGAGAGTGGCAGGCCAC-3′ (forward), 5′-GTGGCCTGCCACTCTCACGTATGTTTCTGGAGG-3′ (reverse); K171R:

5′-GGTACTAAGCAGAGCCAACAGCCAG-3′ (forward), 5′-CTGGCTGTTGGC-TCTGCTTAGTACC-3′ (reverse). Human *ubiquitin* cDNA was cloned into pIRESneo2/FLAG to obtain pIRESneo2/FLAG-ubiquitin.

### Cells

WI38VA13-derived PCNA[WT], PCNA[KR] cells, and control cells harboring empty vector were obtained as described (Kanao et al, 2015a). To obtain RAD18 and Polη KO cells, WI38VA13 cells were transfected with pX459/RAD18 or pX459/Polη, selected with 3 μg/ml puromycin (InvivoGen) for 24 h, and cloned from single colonies. To obtain Polκ KO cells, HeLaS3 cells were transfected with pX459/Polκ, selected with 3 μg/ml puromycin for 24 h, and cloned from single colonies. Mutations in genomic DNA were confirmed by Sanger sequencing. XP2OSSV, CS1ANSV, and CS2OSSV cells were transfected with pMK10, pMK10/PCNAres, or pMK10/PCNA[KR]res and selected with 0.4 mg/ml G418 (Nacalai Tesque). The cells were subsequently transfected with siRNA against endogenous PCNA (siPCNA) to eliminate cells not expressing exogenous PCNA. RAD18 KO cells were transfected with pIREShyg3/FLAG-RAD18 and selected with 0.2 mg/ml hygromycin (FUJIFILM Wako Pure Chemical) to obtain RAD18 KO/FLAG-RAD18 cells. Polη KO cells were trans-fected with pIRESneo2/FLAG-Polη and selected with 0.2 mg/ml G418 to obtain Polη KO/FLAG-Polη cells. To obtain Polκ KO/GFP-Polκ, Polκ KO cells were transfected with pAcGFP/Polκ, and single clones were obtained after selection with 0.2 mg/ml hygromycin. U2OS HLTF KO cells were prepared as described previously (Masuda et al, 2018). WI38VA13 cells were transfected with wild type or mutant pIRESneo2/FLAG-RFWD3 and selected with 0.1 mg/ml G418 to obtain WI38VA13/FLAG-RFWD3 cells. WI38VA13 cells were transfected with wild type or mutant pIRESneo2/FLAG-RPA2 and selected with 0.2 mg/ml G418 to obtain WI38VA13/FLAG-RPA2 cells. WI38VA13, U2OS, XP2OSSV, XP4PASV, CS1ANSV, and CS2OSSV-derived cells were grown in DMEM (FUJIFILM Wako Pure Chemical) supplemented with 10% fetal bovine serum and 1× Penicillin–Streptomycin Mixed Solution (Nacalai Tesque). HeLaS3-derived cells were grown in DMEM supplemented with 5% calf serum and 1× Penicillin–Streptomycin Mixed Solution. BJ1/hTERT cells (Cao et al, 2014) were cultured in Minimum Essential Medium Eagle, Alpha Modification (Sigma-Aldrich) containing 10% fetal bovine serum. HCC1937 and HCC1937+BRCA1 cells (Garcia-Higuera et al, 2001) were cultured in RPMI 1640 (Nacalai Tesque) supplemented with 10% fetal bovine serum and 1× Penicillin–Streptomycin Mixed Solution. PD20F-derived cells (Garcia-Higuera et al, 2001) were grown in DMEM sup-plemented with 15% fetal bovine serum and 1× Penicillin–Streptomycin Mixed Solution. Plasmid transfections were performed using the Neon Transfection System (Thermo Fisher Scientific).

### siRNA experiments

Cells were transfected with the following siRNAs using the Neon Transfection System (Thermo Fisher Scientific) or Dharmafect1 (Horizon Discovery). siPCNA (Kanao et al, 2015a), siREV1 (Akagi et al, 2009), siSHPRH (#M-007167-01), siPolκ pool (#L-021038-00), siPolι (#M-019650-01), siBRCA1#1 (#P-002111-01), siFANCD2 (#L-016376-00), and non-targeting control siRNA (siNTC)#1 (#D-001210-01) were obtained from Dharmacon (Horizon Discovery). siREV7(#S20468), siRFWD3#1 (#S30312), siPolκ#2 (#S28116), siPolη (#s531965),

siBRCA1#2 (#S458), and siNTC#2 (#4390844) were obtained from Thermo Fisher Scientific. siRFWD3#2 (#Hs01_00182002), siRPA2 (#Hs01_00095278), siRAD51 (#Hs01_00096904), and siNTC#3 (#SIC-001) were obtained from Sigma-Aldrich.

## Cellular survival assay

For 3-(4,5-dimethylthiazol-2-yl)-5-(3-carboxymethoxyphenyl)-2-(4-sulfophenyl)-2H-tetrazolium, inner salt (MTS) assays, cells were plated into six-well culture plates, treated with the indicated reagents, and cultured for 4 d. Cellular viability was estimated using CellTiter 96 Aqueous One Solution Cell Proliferation Assay kits (Promega). Colony formation assays were performed as previously described (Kanao et al, 2015a). To replace endogenous with exogenous PCNA, exogenous PCNA–expressing cells were transfected with siRNA against endogenous PCNA and incubated for 3 d before reseeding. Cells were then cultured overnight and treated with illudin S (Bioaustralis), irofulven (Toronto Research chemicals), mitomycin C (MMC) (Nacalai Tesque), camptothecin (CPT) (FUJIFILM Wako Pure Chemical), formaldehyde (FA) (Thermo Fisher Scientific), hydroxyurea (HU) (Sigma-Aldrich), or NU-1025 (Tocris Bioscience). Cells were incubated for 2 d after transfection when siRNAs other than those against PCNA were transfected. Data were represented as mean ± SD from at least three independent experiments.

## Whole-cell lysate preparation and immunoblotting

Whole-cell lysates were prepared as described previously (Kashiwaba et al, 2015), separated by SDS–PAGE, and transferred onto PVDF membranes (Merck Millipore). Membranes were blocked with 5% skim milk (Nacalai Tesque) and incubated with primary antibodies. After washing with TBS (20 mM Tris–HCl [pH 8.0] and 150 mM NaCl) containing 0.1% Tween-20, the membranes were incubated with HRP-conjugated secondary antibodies and sequentially washed in TBS containing 0.1% Tween-20. Signals were detected using ChemiLumi One-L kits (Nacalai Tesque) and an LAS4000 mini system (GE Healthcare). The primary antibodies used were as follows: rabbit anti-RAD18 (1:2,000; 70-023; Bio Academia), mouse anti-PCNA (PC10) (1:10,000 or 1:2,000; sc-56; Santa Cruz Biotechnology), goat anti-Lamin B (C-20) (1:3,000; sc-6216; Santa Cruz Biotechnology), mouse anti-SHPRH (3F8) (1:2,000; TA501443; ORIGENE), rabbit anti-HLTF (1:3,000; ab17984; Abcam), rabbit anti-XPA (FL-273) (1:3,000; sc-853; Santa Cruz Biotechnology), goat anti-CSB (E-18) (1:3,000; sc-10459; Santa Cruz Biotechnology), rabbit anti-XPC (1:2,000) (Sugasawa et al, 1996), guinea pig anti-REV1 (1:10,000) (Akagi et al, 2009), mouse anti-Polκ (A-9) (1:2,000; sc-166667; Santa Cruz Biotechnology), mouse anti-REV7 (1:2,000; 612266; BD Biosciences), rabbit anti-Polι (1:1,000; ab123331; Abcam), mouse anti-FANCD2 (FI17) (1:2,000; sc-20022; Santa Cruz Biotechnology), rabbit anti-BRCA1 (C20) (1:3,000; sc-642; Santa Cruz Biotechnology), mouse anti-RPA2 (1:2,000; ab2175; Abcam), mouse anti-γH2AX (1:2,000; 05-636; Merck Millipore), rabbit anti-phospho RPA2 (S4/S8) (1:2,000; A300-245A; Bethyl Laboratories), rabbit anti-H2AX (1:2,000; ab11175; Abcam), rabbit anti-Lamin B1 (1:10,000; 12987-1-AP; Proteintech), mouse anti-β-actin (6D) (1:10,000; M177-3; Medical & Biological Laboratories), and rabbit anti-ubiquityl PCNA (Lys164) (D5C7P) (1:1,000; 13439; Cell Signaling Technology). Rabbit polyclonal anti-Polη antibody (1:20,000) was raised against the C-terminal region (aa 507–713) of human Polη (Medical & Biological Laboratories). Rabbit polyclonal anti-RFWD3 antibody (1:2,000) was raised using a synthetic peptide (aa 187–204) of human RFWD3 (Sigma-Aldrich). The following secondary antibodies were used: anti-mouse IgG-HRP (1:3,000; 330; Medical & Biological Laboratories), anti-rabbit IgG-HRP (1:3,000; 458; Medical & Biological Laboratories), anti-goat IgG-HRP (1:3,000; 546; Medical & Biological Laboratories), and anti-Guinea Pig IgG (1:5,000; Chemicon; Merck Millipore).

## BrdU incorporation analysis

For the pulse-chase assay, cells were treated simultaneously with 20 μM BrdU (BD Biosciences) and 25 ng/ml illudin S for 1 h, and then incubated for the indicated periods without the drugs. For pulse-labeling assays, cells were treated with 25 ng/ml illudin S for 1 h, incubated without the drug for the indicated periods, and then treated with 20 μM BrdU for 1 h. Cells were harvested and fixed with 70% ethanol at −20°C. The fixed cells were treated with 2 N HCl supplemented with 0.5% Triton X-100. After washing twice with 0.1 M Na$_2$B$_4$O$_7$ and once with dilution buffer (1% BSA and 0.1% Tween-20 in PBS), the cells were incubated with Alexa Fluor 488–conjugated anti-BrdU antibody (BioLegend). After treatment with 0.2 mg/ml RNase A, DNA was stained with 7-amino-actinomycin D (7-AAD) (Beckman Coulter). Data were collected on an FC500 flow cytometer (Beckman Coulter) and analyzed using FlowJo software (BD Biosciences). Proportions of BrdU-positive cells were calculated by gating using FlowJo software.

## RPA and γH2AX detection

Cells were treated with 25 ng/ml illudin S for 1 h and incubated for the indicated times without the drug. Detergent-soluble materials were eliminated by incubation with 0.5% Triton X-100 in PBS on ice for 5 min. The cells were then fixed with 3.7% formaldehyde in PBS for 15 min at room temperature. The fixed cells were blocked with 3% BSA in PBS for 30 min at room temperature and stained using mouse anti-RPA2 (ab2175; Abcam) and rabbit anti-γH2AX (9718; Cell Signaling Technology). RPA and γH2AX signals were visualized with Alexa Fluor 488–conjugated anti-mouse antibody (Thermo Fisher Scientific) and Alexa Fluor 594-conjugated anti-rabbit antibody (Thermo Fisher Scientific), respectively. Nuclei were stained with 2 μg/ml Hoechst 33342. Images were collected using an LSM710 confocal microscope with Plan-Apochromat 40×/1.3 NA Oil (Zeiss). Fluorescence intensities were quantified using Zeiss Zen software.

## DNA fiber assay

Cells were labeled with 25 μM 5-Chloro-2′-deoxyuridine (CldU) (Sigma-Aldrich) for 30 min, treated with 50 ng/ml illudin S for 1 h, incubated for 0, 1, or 3 h without illudin S, and labeled with 250 μM 5-Iodo-2′-deoxyuridine (IdU) (Tokyo Chemical Industry) for 30 min. To test DNA synthesis after UV irradiation, CldU-labeled cells were irradiated with 8 J/m$^2$ UV, then labeled with IdU for 30 min. The labeled cells were collected and resuspended in PBS to reach a concentration of 5 × 10$^5$ cells/ml. Next, 2.5 μl of cell suspension was mixed with 7.5 μl of lysis buffer (200 mM Tris–HCl [pH 7.5], 50 mM EDTA, 0.5% SDS) on glass slides (Matsunami Glass) and incubated for 8 min. The slides were tilted 10–15° for DNA spread, air-dried, and fixed in 3:1 methanol/acetic acid. After denaturing with 2.5 M HCl for 1 h, DNA fibers

were blocked with 2% BSA in PBST (1× PBS, 0.05% Tween-20) for 30–40 min and stained using rat anti-BrdU (CldU) BU1/75 (ICR1) (1:100; ab6326; Abcam) and mouse anti-BrdU (IdU) B44 (1:40; 347580; BD Biosciences) for 2.5 h in the dark. CldU and IdU signals were visualized with Alexa Fluor 594-conjugated anti-rat antibody (Thermo Fisher Scientific) and Alexa Fluor 488-conjugated anti-mouse antibody (Thermo Fisher Scientific), respectively. Images were collected using an LSM710 confocal microscope with Plan-Apochromat 100×/1.40 NA Oil (Zeiss). Track length of DNA fibers was quantified using ImageJ software (National Institute of Health).

### siRNA library screening

BJ1/hTERT (hTERT-immortalized normal human foreskin fibroblast) cells were seeded in 384-well plates (CellCarrier-384 Ultra; PerkinElmer) and transfected with siRNAs (silencer select, pool of three siRNAs per one gene, final siRNA concentration: 3 nM per well; Thermo Fisher Scientific) using Lipofectamine RNAiMAX Transfection Reagent (Thermo Fisher Scientific). 24 h after transfection, cells were treated with either: (1) 2 µg/ml irofulven for 1 h and cultured for another 4 d without irofulven or (2) with 75 ng/ml irofulven for 4 d. Cells were fixed with 4% paraformaldehyde and stained with 1 µg/ml Hoechst 33342 (Thermo Fisher Scientific) in PBS. The proportions of stained cells were determined using an Opera Phenix system and Harmony software (PerkinElmer).

### FLAG-Ub immunoprecipitation assays

For RFWD3 depletion, WI38VA13 cells were transfected with non-targeting control siRNA or the siRNA targeting RFWD3 and cultured for 2 d. The cells were transfected with pIRESneo2/FLAG-Ub. 24 h after transfection with the plasmids, the cells were treated with 50 ng/ml illudin S for 1 h and incubated for the indicated times without the drug. Whole-cell extracts were prepared by resuspending cells in RIPA buffer (20 mM Tris–HCl [pH 7.6], 150 mM NaCl, 10% glycerol, 0.1 mM EDTA, 1% Triton X-100, 1% deoxycholate, 0.1% SDS, 0.25 mM phenylmethylsulfonyl fluoride, Complete Protease Inhibitor Cocktail [Merck], and PhosSTOP [Merck]) and centrifuging at 4°C. The supernatants were incubated with anti-FLAG affinity gel (Sigma-Aldrich) at 4°C. After washing the beads three times with RIPA buffer, the precipitated proteins were eluted with RIPA buffer containing 0.2 mg/ml FLAG peptide (Sigma-Aldrich). The extracts and precipitated proteins were analyzed by immunoblotting using the following antibodies: rabbit anti-RPA2 (1:2,000; A300-244A; Bethyl Laboratories), rabbit anti-RAD51 (1:1,000; 70-001; Bio Academia), anti-rabbit IgG-HRP (1:3,000; 458; Medical & Biological Laboratories), and anti-FLAG-HRP (1:5,000; A8592; Sigma-Aldrich).

### Cell fractionation

Cell fractionation was conducted as previously described (Vujanovic et al, 2017), with modifications. Briefly, cells were treated with 200 ng/ml (WI38VA13) or 100 ng/ml (U2OS) illudin S or irradiated with 20 J/m$^2$ UV-C following incubation for 3 or 6 h. Collected cells were resuspended in SB1 buffer (50 mM Tris–HCl [pH 8.0], 150 mM NaCl, 1 mM EDTA, 0.1% Triton X-100, 1 mM phenylmethylsulfonyl fluoride, Complete Protease Inhibitor Cocktail [Merck], PhosSTOP [Merck], and

2 mM N-ethylmaleimide [NEM; FUJIFILM Wako Pure Chemical]) and centrifuged at 600$g$ for 5 min at 4°C. This extraction was performed twice. The pellet was resuspended into SB2 buffer (50 mM Tris–HCl [pH 8.0], 1 mM MgCl$_2$, 1 mM phenylmethylsulfonyl fluoride, PhosSTOP, and 2 mM NEM), sonicated, incubated with benzonase (EMD Millipore) at 30°C for 30 min, and centrifuged at 600$g$ for 5 min at 4°C. The pellet was resuspended in SB2 buffer and centrifuged at 4°C. The resulting supernatants were collected as the chromatin fraction and used for immunoblotting analyses.

### Statistical analysis

GraphPad Prism 8 software (GraphPad Software) was used for statistical analysis. Two-tailed, unpaired $t$ tests were used to analyze survival assays, and two-tailed, unpaired Welch's $t$ tests were used for all other comparisons.

## Supplementary Information

## Acknowledgements

We are grateful to Dr. Yuji Masuda for his critical reading of the manuscript and helpful comments. We also thank Mie Takahashi, Yukiko Takeuchi, and Asako Nishikawa for their assistance with laboratory work. This work was supported by JSPS KAKENHI (21K12238 to R Kanao and 20H04335, 21K19843 to C Masutani) and the Takeda Science Foundation (to R Kanao and C Masutani). Part of this study was conducted through the Joint Usage/Research Center Program of the Radiation Biology Center, Kyoto University, and through the same program at the Research Institute for Radiation Biology and Medicine (RIRBM), Hiroshima University.

### Author Contributions

R Kanao: conceptualization, funding acquisition, investigation, and writing—original draft.
H Kawai: investigation and writing—review and editing.
T Taniguchi: resources and writing—review and editing.
M Takata: resources and writing—review and editing.
C Masutani: conceptualization, funding acquisition, writing—original draft, and project administration.

### Conflict of Interest Statement

The authors declare that they have no conflict of interest.

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
