## [Reviewer comments · Life Science Alliance]

Life Science Alliance

RFWD3 and translesion DNA polymerases contribute to PCNA modification-dependent DNA damage tolerance

Rie Kanao, Hidehiko Kawai, Toshiyasu Taniguchi, Minoru Takata, and Chikahide Masutani

DOI: <https://doi.org/10.26508/lsa.202201584>

Corresponding author(s): Chikahide Masutani, Nagoya University

Review Timeline:

Submission Date:	2022-06-29
Editorial Decision:	2022-06-30
Revision Received:	2022-07-01
Editorial Decision:	2022-07-01
Revision Received:	2022-07-03
Accepted:	2022-07-06

Transaction Report:

Please note that the manuscript was previously reviewed at another journal and the reports were taken into account in the decision-making process at Life Science Alliance.

Reviewer #1 Review

Comments to the Authors (Required):

In this revised manuscript, Kanao et al. have investigated the role of the TLS pathway in mediating tolerance of illudin S /irofulven-induced DNA damage. These workers have incorporated many new experiments and addressed many of the points addressed by this reviewer. The revised manuscript is definitely improved by the addition of DNA fiber assays to determine the effects of genotoxin treatments / genetic ablations on DNA replication, inclusion of appropriate controls for knockdowns, and more clear explanations of some of the technical aspects (e.g. replacing endogenous PCNA with ubiquitination-resistant mutants).

The overall conclusion is that unidentified RFWD3 targets are required for Illudin S-tolerance via a genome maintenance sub-pathway that is separable from the Y-family TLS -polymerases. This conclusion is fully supported by a large body of original and new data.

The strengths are that this is a detailed and meticulous study and the data showing separation of RFWD3 and Y-family polymerase-mediated pathways are convincing and interesting.

However, THE major limitations of the study, as explicitly stated in the original review remain: There is no mechanistic connection between monoubiquitinated PCNA-Ub and RFWD3, nor any mechanistic insight into relevant targets or pathways downstream of RFWD3.

Moreover, Polk was previously identified as an important factor for tolerating Illudin S (Olivieri et al., Cell 2020) and therefore the role of Polk in this response is not novel.

Minor comment:

P8 - effect of Polk depletion on illudin S resistance was smaller than that of replacing endogenous PCNA with unmodifiable PCNA (Fig. 3A), prompting us to search for additional factors involved in illudin S and irofulven tolerance

Could this not simply indicate that Polk is not completely ablated (siRNA) and/or that Polk regulated to some extent by PCNA-Ub-independent mechanisms?

Reviewer #2 Review

Comments to the Authors (Required):

The manuscript from Kanao et al. has been significantly improved by addressing many of the reviewers' comments. Particularly, they have complemented their original S phase progression via FACS by DNA fiber analysis to directly monitor DNA replication, which convincingly show that Polk and RFWD3 are required for DNA synthesis upon illudin S-treatment, which is known to induce minor groove DNA lesions. They also show via knock down rescue experiment that RPA2 ubiquitylation is not required for this role of RFWD3 in DDT suggesting that this function of RFWD3 is HR independent. However, RPA-RFWD3 interaction is important, suggesting that RPA recruits RFWD3 to perform its function. According to reviewers' comments, they also monitored PCNA ubiquitylation via chromatin fractionation and report a small defect in PCNA ubiquitylation in some of their cell lines.

Although the manuscript has been significantly improved and show many experiments that are worth publishing, I am still uncertain on the main conclusions and advancement of this study. The 2 main findings are:

1. that Polk participates in illudin-S tolerance, which was previously known based on sensitivity screen and the fact that Polk is the only polymerase known to replicate across minor groove lesions (which is what illudin S generates).
2. RFWD3 participates in illudin-S tolerance, which strongly suggests a role in DDT, which was previously shown by Gallina et al. 2021. This manuscript extends these original findings by showing that RFWD3 is critical for DNA synthesis across minor groove lesions by DNA fiber analysis.

Moreover, although the conclusions that RFWD3 and Polk participate in independent pathways have been toned down, this idea is still too much enforced in my opinion without experiments substantiating it. Particularly, the authors do not know how RFWD3 functions in their system and calling this a "new branch" regulated by RFWD3 is a misleading statement. If this manuscript was to be accepted, I still think the conclusions should be further toned down significantly. One clear example is in the abstract:

"RFWD3 contributes to a novel branch of PCNA modification dependent DNA damage tolerance". It is clear from their screen and other screens that REV3 (Polzeta) is also essential for illudin S resistance. Thus, the role of RFWD3 could be upstream and regulate all TLS events (including Polk and Rev3), which was proposed and would be consistent with Gallina et al.

Reviewer #1 Review

Comments to the Authors (Required):

Kanao et al. have investigated the role of the TLS pathway in mediating tolerance of illudin S / irifolven-induced DNA damage. These workers show that RAD18 and PCNA mono-ubiquitination are required for tolerance of illudin S / irifolven genotoxicity. However, the Y-family DNA Polymerases downstream of PCNA-Ub are either dispensable (POLH, REV1, REV7, POLI) or are less essential than PCNA mono-ubiquitination (POLK) for DNA damage tolerance. The investigators perform a DNA repair-focused siRNA screen for alternative factors that confer tolerance of illudin S-induced DNA damage. RFWD3 (aka FANCW) emerges as an important gene for Illudin S / irifolven-tolerance, as also reported by Olivieri et al. 2020. Published RFWD3 mutants lacking E3 ubiquitin ligase activity or chromatin/RPA-interaction fail to complement DNA damage-sensitivity of RFWD3-depleted cells. RPA is a known substrate for RFWD3. However, ubiquitination-resistant RPA mutants fail to phenocopy RFWD3-deficiency, indicating that unidentified RFWD3 substrates are required for Illudin S-tolerance. The overall conclusion is that RFWD3 contributes to DNA damage tolerance and S-phase progression via a sub-pathway of that is separable from the Y-family TLS -polymerases.

The strengths are that this is a detailed and careful study and the data showing separation of RFWD3 and Y-family polymerase-mediated pathways are interesting. However, major limitations of the study are that there is no mechanistic connection between monoubiquitinated PCNA-Ub and RFWD3, nor any mechanistic insight into relevant targets or pathways downstream of RFWD3. Therefore, in its current form the manuscript is lacking and does not significantly advance our current understanding of how RFWD3 confers Illudin S-tolerance. Other comments and suggestions are listed below.

Comments

1. BrdU/FACS assays lack the resolution needed to determine roles of RFWD3 in regulating DNA replication. DNA replication dynamics would be best analyzed by DNA fiber assays which yield important information on initiation and elongation events, origin firing etc. Given the proposed role of RFWD3 in DNA damage tolerance it would also very helpful to determine whether newly synthesized DNA strands in RFWD3-depleted cells are short and discontinuous (hallmarks of TLS) after UV vs illudin S treatment.

2. The suggestion that RFDW3 pathway is downstream of PCNA ubiquitination is based solely on epistasis analyses. How is PCNA monoubiquitination regulating RFDW3?
3. The known function of the RFDW3 and RPA interaction is to promote HR. What is the contribution of the HR pathway to RFDW3-mediated DNA damage tolerance under these experimental conditions?
4. It is not clear from the methods presented whether the researchers used the siPCNA to knockdown the endogenous PCNA of the PCNA[WT] and PCNA[KR] cells when they did the survival experiments. Details of the PCNA replacement experiments should be included in this manuscript.
5. In Fig. S2, siRAD18 should be included in both A and C. Without the siRAD18 positive control it is not possible to conclude whether PCNA modification impacts illudin S-sensitivity in U2OS cells.
6. What is the ranking of RFDW3 in the siRNA screen? What about all the other genes listed in Table S1?
7. Fig 5C is missing a very important control, namely vec+siRPA. The authors must show how RPA-depletion under these conditions affects illudin C-sensitivity.
8. The authors state 'RFDW3- and Polk-mediated DNA damage tolerance pathways are independent of each other, but both depend on PCNA modifications at K164'. The experiments showing dependence of RFDW3 and POLK -mediated DNA damage tolerance pathways on PCNA modifications at K164 are performed using WI38VA13 cell lines (over-expressing either wild-type or K164R (KR) mutant exogenous PCNA). However, RFDW3 and POLK co-depletion experiments (showing that RFDW3 and POLK function in independent pathways and that they are solely responsible for PCNA-Ub-mediated damage tolerance) were performed with HeLaS3 cells.

When comparing effects of the PCNA K164R (KR) mutant (Fig. 1A), RAD18 deletion (Fig. 1E), Polk depletion (Fig. 4A) and RFDW3 depletion (Fig. 4E) on illudin-S sensitivity in WI38VA13 cells, there are no significant differences for K164R (KR) mutant, RAD18 deletion, and RFDW3 depletion. Therefore, a co-depletion experiment in WI38VA13 (similar to Fig 7A examining UV-induced damage) is necessary to test whether RFDW3 and POLK function in independent pathways.
9. The impact of RFDW3-depletion on PCNA ubiquitylation and POLK levels is assessed only in WI38VA13 cells. Furthermore, these effects are not quantified and are hard to interpret. The effect of RFDW3-depletion on PCNA ubiquitylation, RAD18, POLK, and POLH levels following Illudin-S or UV-treatment should also be determined in other cell lines. This is an important issue given the recent study by Gallina et al. showing that RFDW3 promotes TLS by stimulating PCNA ubiquitylation.
10. Page 8, second paragraph "Together with the result that the RFDW3 I639K mutant was not able to complement illudin S-sensitivity, these results indicate that the interaction between RFDW3 and RPA is required for cell survival after illudin S treatment, and is required during ICL repair." There are no data to show that RFDW3/ RPA interaction is required during ICL repair in this paper. If this is a result from another study, please include the relevant citation.
11. In Fig 7B what is the phenotype of a POLH and RFDW3 double knockdown?
12. In Fig 4H what is the phenotype of a POLK and RFDW3 double knockdown?

Reviewer #2 Review

Comments to the Authors (Required):

Kanao et al. present an interesting study where they highlight the role of RFDW3 and Polk in the response to Illudin S treatment. Via a combination of genetic experiments, they conclude that both Polk and RFDW3 participate in DNA damage tolerance downstream of PCNA ubiquitylation. Although the experiments are generally well performed and clearly presented, I am concerned about the interpretation of the data as the authors might be misled by the inability to detect PCNA poly-ubiquitylation. The following major and minor points should be addressed to envision publication.

Major points

- The conclusion that PCNA mono but not poly-ubiquitylation is essential for Illudin S response is key to the manuscript but unfortunately not properly addressed as it is very difficult to monitor PCNA poly-ubiquitylation in human cells and likely impossible to detect it via whole cell extracts as was done here. One should either do a proper chromatin fractionation (Vujanovic et al. Molecular Cell 2017) or a PCNA pulldown under denaturing conditions (Gallina et al. Molecular Cell 2021). A positive control known to induce PCNA poly-ubiquitylation should be added and directly compared to Illudin S to back up their conclusions (i.e. UV).

Although the authors address the role of PCNA poly-ubiquitylation by monitoring the Illudin S sensitivity of HLTf and SHPRH DKO, recent data suggest that these enzymes are not essential for PCNA poly-ubiquitylation (Krijger et al. DNA repair 2011; Gallina et al. Molecular Cell 2021) and should therefore not be taken as such unless a PCNA poly-ubiquitylation blot in HLTf-SHPRH DKO clearly shows the absence of PCNA poly-ubiquitylation.

- The sensitivity of PolK, RAD18 and RFWD3-deficient cells to Illudin S was already reported and the reference should be included to the manuscript (Olivieri et al. Cell 2020). Note that in that study, REV1, REV3L and REV7 were also shown to be sensitive to Illudin S, which contrasts to the results presented here. The discrepancy might be caused by the incomplete ablation of REV1, REV3L or REV7 via siRNA in this study and should therefore be acknowledged in the manuscript.
- Although previously published, it is important to show a new PCNA blot of WI38VA13 cells expressing either WT or KR mutant (since the entire manuscript relies on this system).
- Results of the screen which identified RFWD3 as a candidate must be presented in the manuscript (table S1) so that we can compare RFWD3 to other candidate genes.
- Gallina et al. Molecular Cell 2021 was published over 6 months ago and should therefore be included in the introduction of the manuscript since the findings reported in that manuscript are highly relevant for the current study. Importantly, Gallina et al. showed that RFWD3 is essential for the bypass of a wide range of DNA lesions (ICLs, DPCs and UV-CPDs) and not just DPCs as mentioned in the discussion by the authors. Thus, the effect of RFWD3 regulating TLS can be extrapolated to different lesions and could align with the hypersensitivity of RFWD3-depleted cells to illudin S treatment.
- Furthermore, Gallina et al. showed that PCNA mono-ubiquitylation during DNA replication is unaffected in the absence of RFWD3 but instead RFWD3 depletion abolishes damaged induced PCNA ubiquitylation (both in *Xenopus* egg extracts and in human cells). The results presented here and in Figure S6C could be in accordance with Gallina et al. and one possibility is that in the absence of RFWD3 and illudin S treatment, PCNA poly-ubiquitylation is abolished leading to the absence of DNA damage tolerance (Gallina et al. work suggests that a branch of TLS is dependent on PCNA poly-ubiquitylation). To clearly define the role of RFWD3 in illudin-S dependent damage tolerance, the authors need to also monitor PCNA poly-ubiquitylation as suggested above in the presence or absence of RFWD3.

Similarly, the genetic interaction between PolK and RFWD3 presented in Figure 6 would suggest that they play independent roles to mediate DNA damage tolerance. Alternatively, one possibility is that RFWD3 plays a role upstream in the pathway controlling the different branches of DTT via PCNA polyubiquitylation while PolK only participates to a subset of TLS. The use of siRNA against RFWD3 (instead of true KO) might be the reason of the additive effect seen when both RFWD3 and PolK are depleted.

Note that the role of RFWD3 in UV-lesion induced bypass was previously addressed in Gallina et al. but not referenced.

- The general title is a bit misleading since the study is quite specific to illudin S, PolK and RFWD3. Maybe "RFWD3 and PolK contribute to PCNA ubiquitylation dependent resistance to illudin S" would be more appropriate.

Minor points

- Page 2
"pol eta-deficient cells are more sensitive to cisplatin treatment" -More sensitive than what?
- Page 4
"Replacement of PCNA with mutant PCNA[KR] did not increase cellular sensitivity to mitomycin C (MMC), camptothecin (CPT), formaldehyde (FA), hydroxyurea (HU), or the PARP inhibitor NU-1025 (Fig. S1), indicating that PCNA modifications are not required for tolerance to these compounds."

These conclusions need to be toned down since this is in the background of siPCNA and the authors cannot discard the possibility that residual endogenous PCNA contributes to the resistance of these cells.

- Page 4
Typo-Illudin S
- Figure 2
Panels A, B and C should be combined to another figure (i.e. Figure 1) or placed in supplemental since they deviate from the main focus of the manuscript and do not justify a figure on their own.
- A more detailed introduction on DDT including the mechanism of template switching might be helpful to the reader.

Reviewer #3 Review

Comments to the Authors (Required):

Kanao and collaborators describe an interesting, careful and convincing study analyzing the mechanism of action of alkylating agent Illudin S. This and related compounds are being explored as chemotherapeutic drugs, so it is important to learn how cells handle these adducts.

In an incisive drug screen using PCNA-replacement K164R cells, the authors find that Illudin S sensitivity involves a mechanism of tolerance that is pol eta independent, yet dependent on PCNA modification (ubiquitination at K164R).

S phase progression is delayed by exposure to Illudin S, when K164 ubiquitination of PCNA is not possible. To analyze the mechanism needed to overcome such blocks to replication, the authors ruled out three E3 ligases. They also found that much of the sensitivity is independent of NER processes. The complementation experiments throughout the manuscript are excellent and convincing.

In an siRNA screen the authors found two factors that help in overcoming Illudin S-induced DNA replication blockage, pol kappa and the E3 ligase RFWD3. These factors acting independent of one another. These are important results that should be reported.

Although the mechanism of RFWD3 action is not completely understood, the authors show that it is not due to RPA ubiquitination but involves RPA interaction. It is independent of FANCD2, even though RFWD3 is thought to be a FANCD gene. The observed UV sensitivity of RFWD3 cells is another important new observation which suggests that the gene might be more relevant to other sensitivity disorders.

The paper is generally well written and described, I suggest a couple of clarifying changes in the Introduction:

1. Page 3 ERCC1-defective cells deficient in BER? - this should be XRCC1 (c.f. Jaspers et al 2002), x-ray cross complementation group 1
2. The GG-NER discussion could be clearer regarding the Jaspers et al 2002 results. The challenge in describing this is that XP-A cells (for example) are defective in both GG-NER and TC-NER. I suggest "Cells that are GG-NER deficient but TCR-proficient are only as sensitive to these compounds as ..."

June 30, 2022

Re: Life Science Alliance manuscript #LSA-2022-01584-T

Prof Chikahide Masutani
Nagoya University
Department of Genome Dynamics
Research Institute of Environmental Medicine
Furo-cho, Chikusa
Nagoya, Aichi 464-8601
Japan

Dear Dr. Masutani,

Thank you for submitting your manuscript entitled "RFWD3 and DNA polymerase κ contribute to PCNA ubiquitination-dependent resistance to illudin S" to Life Science Alliance. We invite you to submit a revised manuscript addressing Reviewer 2's remaining requests to further tone down certain conclusions.

Thank you for this interesting contribution to Life Science Alliance. We are looking forward to receiving your revised manuscript.

Sincerely,

Eric Sawey, PhD
Executive Editor
Life Science Alliance
<http://www.lsa-journal.org>

B. MANUSCRIPT ORGANIZATION AND FORMATTING:

Thank you for inviting us to submit our revised manuscript, now entitled “RFWD3 and translesion DNA polymerases contribute to PCNA modification-dependent DNA damage tolerance”. Following reviewer 2’s remaining requests to further tone down certain conclusions, we have improved the manuscript, which we hope is now suitable for publication in your esteemed journal.

Our responses to the reviewer 2’s remaining concerns are as follows (reviewer comments are italicized):

Although the manuscript has been significantly improved and show many experiments that are worth publishing, I am still uncertain on the main conclusions and advancement of this study. The 2 main findings are:

- 1. that Polk participates in illudin-S tolerance, which was previously known based on sensitivity screen and the fact that Polk is the only polymerase known to replicate across minor groove lesions (which is what illudin S generates).*
- 2. RFWD3 participates in illudin-S tolerance, which strongly suggests a role in DDT, which was previously shown by Gallina et al. 2021. This manuscript extends these original findings by showing that RFWD3 is critical for DNA synthesis across minor groove lesions by DNA fiber analysis.*

Moreover, although the conclusions that RFWD3 and Polk participate in independent pathways have been toned down, this idea is still too much enforced in my opinion without experiments substantiating it. Particularly, the authors do not know how RFWD3 functions in their system and calling this a "new branch" regulated by RFWD3 is a misleading statement. If this manuscript was to be accepted, I still think the conclusions should be further toned down significantly. One clear example is in the abstract:

*"RFWD3 contributes to a novel branch of PCNA modification dependent DNA damage tolerance".
It*

is clear from their screen and other screens that REV3 (Polzeta) is also essential for illudin S resistance. Thus, the role of RFWD3 could be upstream and regulate all TLS events (including Polk and Rev3), which was proposed and would be consistent with Gallina et al.

According to the reviewer's comment, we toned down the descriptions about the relation of RFWD3 and Pol κ , mainly by removing the words "novel branch" and "independently", as follows.

The last sentence of the abstract pointed out by the reviewer "RFWD3 contributes to a novel branch of PCNA-modification dependent DNA damage tolerance" is replaced by "RFWD3 contributes to PCNA-modification dependent DNA damage tolerance in addition to translesion DNA polymerases". We also removed the word "independently" from a sentence in the abstract "Polk and RING finger and WD repeat domain 3 (RFWD3) independently contribute to tolerance, but are both dependent on PCNA modifications", which is now "Polk and RING finger and WD repeat domain 3 (RFWD3) contribute to tolerance, and are both dependent on PCNA modifications" in the revised manuscript.

Sentences in the last paragraph of the Introduction are also modified as follows. The sentence "we found that human Polk and RFWD3 contribute to overcoming replication arrest independently of each other, but are dependent on PCNA modification at K164" is replaced by "we found that human Polk and RFWD3 contribute to overcoming replication arrest dependently on PCNA modification at K164."

We also replaced the last sentence of the results "Our results suggest that PCNA modifications at K164 generally contribute to two branches of DNA damage tolerance—one involving RFWD3, and the other involving a TLS polymerase appropriate for the type of DNA lesion (Fig. 6G)" by "Our results suggest that PCNA modifications at K164 generally contribute to DNA damage tolerance involving RFWD3 and TLS polymerases appropriate for the type of DNA lesion (Fig. 6G)."

We also removed "independently" from the Summary blurb which is now "We demonstrate that RING finger and WD repeat domain 3 (RFWD3) and the translesion DNA polymerases Polk and Pol η contribute to PCNA modification-dependent DNA damage tolerance in human cells."

We also agree that from our screen and other screens REV3 (Polzeta) also contributes to illudin S resistance in addition to Polkappa. In addition, RFWD3 contributes to UV resistance in addition to Poleta. We also agree "RFWD3 could be upstream and regulate all TLS events". Considering the involvement of RFWD3 and multiple TLS polymerases in PCNA ubiquitination-dependent tolerance, we changed the title from "RFWD3 and DNA polymerase κ contribute to PCNA ubiquitination–

dependent resistance to illudin S” to “RFWD3 and translesion DNA polymerases contribute to PCNA modification–dependent DNA damage tolerance”.

We appreciate all the comments from the reviewers.

We thank you for considering our revised manuscript for publication in the *Life Science Alliance*.

July 1, 2022

RE: Life Science Alliance Manuscript #LSA-2022-01584-TR

Prof. Chikahide Masutani
Nagoya University
Department of Genome Dynamics
Research Institute of Environmental Medicine
Furo-cho, Chikusa
Nagoya, Aichi 464-8601
Japan

Dear Dr. Masutani,

Thank you for submitting your revised manuscript entitled "RFWD3 and translesion DNA polymerases contribute to PCNA modification-dependent DNA damage tolerance". We would be happy to publish your paper in Life Science Alliance pending final revisions necessary to meet our formatting guidelines.

- please add an alternate abstract / summary blurb, category, to our system
- please add the Twitter handle of your host institute/organization as well as your own or/and one of the authors in our system
- please add a conflict of interest statement to the main manuscript
- please use the [10 author names, et al.] format in your references (i.e. limit the author names to the first 10)
- please add the supplemental figure legends to the main manuscript text
- we encourage you to introduce your panels in your figure legends in alphabetical order

A. FINAL FILES:

B. MANUSCRIPT ORGANIZATION AND FORMATTING:

Sincerely,

July 6, 2022

RE: Life Science Alliance Manuscript #LSA-2022-01584-TRR

Prof. Chikahide Masutani
Nagoya University
Department of Genome Dynamics
Research Institute of Environmental Medicine
Furo-cho, Chikusa
Nagoya, Aichi 464-8601
Japan

Dear Dr. Masutani,

Thank you for submitting your Research Article entitled "RFWD3 and translesion DNA polymerases contribute to PCNA modification-dependent DNA damage tolerance". It is a pleasure to let you know that your manuscript is now accepted for publication in Life Science Alliance. Congratulations on this interesting work.

DISTRIBUTION OF MATERIALS:

Again, congratulations on a very nice paper. I hope you found the review process to be constructive and are pleased with how the manuscript was handled editorially. We look forward to future exciting submissions from your lab.

Sincerely,
